# Investigation of Thermodynamic Properties of Dimethyl Phosphate-Based ILs for Use as Working Fluids in Absorption Refrigeration Technology

**DOI:** 10.3390/molecules28041940

**Published:** 2023-02-17

**Authors:** Michał Skonieczny, Marta Królikowska, Marek Królikowski

**Affiliations:** 1Doctoral School, Warsaw University of Technology, Plac Politechniki 1, 00-661 Warsaw, Poland; 2Department of Physical Chemistry, Faculty of Chemistry, Warsaw University of Technology, Noakowskiego 3, 00-664 Warsaw, Poland

**Keywords:** ionic liquid, ethanol, VLE, density, viscosity, correlation, absorption refrigeration

## Abstract

In the current research, the binary solution containing ionic liquid (IL), 1-ethyl-1-methylmorpholinium dimethyl phosphate ([C_1_C_2_MOR][DMP]), 1-ethyl-1-methylpiperidinium dimethyl phosphate ([C_1_C_2_PIP][DMP]), or *N*,*N*,*N*-triethyl-*N*-methylammonium dimethyl phosphate ([N_1,2,2,2_][DMP]) with ethanol are investigated as new working fluids for absorption refrigeration technology. The IL was mixed with ethanol, which was considered as a refrigerant. Experimental (vapor + liquid) phase equilibria (VLE) of these binary systems were measured by an ebulliometric method within a temperature range from *T* = (328.15 to 348.15) K with an increment of 10 K and pressures up to 90 kPa. Experimental VLE data were correlated using non-random two-liquid (NRTL) within the maximum average relative deviation of 0.45%, which confirms the effectiveness of using such a model for calculations. Each of the proposed binary systems exhibit a negative deviation from Raoult’s law, which is a very important characteristic for working pairs used in absorption heat pumps or absorption refrigerators. From a technological point of view, measurements of physicochemical properties play an important role. In this research, liquid density and dynamic viscosity were determined at temperatures from *T* = (293.15 to 338.15) K at ambient pressure over the whole concentration range. These properties were correlated using empirical equations. From experimental density data, the excess molar volumes were determined and correlated using the Redlich–Kister type equation. Ionic liquid: [C_1_C_2_MOR][DMP] and [C_1_C_2_PIP][DMP] were synthesized and characterized using NMR analysis. The thermophysical characterization of pure ILs, including glass transition temperature (*T*_g_) and heat capacity at the glass transition temperature (Δ_g_*C*_p_), was determined using the differential scanning calorimetry technique (*DSC*) at atmospheric pressure. In this work, the combination of basic studies on the effect of the cation structure of an ionic liquid on the properties of their solutions with ethanol and the possibility of future application of the tested systems in a viable refrigeration system are presented.

## 1. Introduction

With the need to conserve energy resources and protect the environment, the interest in absorption refrigeration equipment as an alternative to compressor units has grown significantly in recent years.

Absorption chillers are devices that make extensive use of thermal energy to produce cooling. These devices, using the so-called thermal compressor (absorber, pump, desorber system) for the process of compressing refrigerant vapors, are an ideal economic solution for unearned surplus technological or waste heat. They also work well where there is a shortage of electricity. Unlike compressor chillers, the generation of cooling in absorption devices does not require the supply of expensive electricity because the energy supply of the absorption device is heat in the form of hot water or steam (so-called waste heat). An additional advantage of absorption devices is the use of an environmentally friendly refrigerant that does not destroy the ozone layer (water). This type of equipment is most often used in industry, especially in the energy-heating sector, because large amounts of waste and inexpensive thermal energy are available in this sector.

The characteristic feature that distinguishes absorption equipment from classical compressor equipment is the existence of the following two internal circuits: the thermal compressor circuit, where the working medium is a solution of absorbent (liquid absorbing substance) and absorbate (gaseous absorbed substance, which is also a refrigerant), and the actual refrigeration circuit, where the working medium is the absorbent itself. One of the most important factors determining the effectiveness of absorption chillers is the properties of the working fluids. In absorption refrigeration, a volatile substance acts as the refrigerant (absorbent), while the absorbent is a less volatile compound that has a strong affinity for the refrigerant. Working fluids should meet the following requirements: (1) the difference in boiling point between the pure refrigerant and the mixture at the same pressure condition should be as large as possible; (2) refrigerant should exhibit high heat of vaporization and high concentration within the absorbent to maintain a low circulation rate between the generator and the absorber per unit cooling capacity; (3) transport properties that influence heat and mass transfer, such as density, viscosity, thermal conductivity, and diffusion coefficient, should be favorable; (4) both refrigerant and absorbent should be non-corrosive, environmentally friendly, and low cost [1].

At present, the most used in refrigeration technology are the following two types of absorption chillers: bromolite (LiBr + H_2_O) and ammonia (H_2_O + NH_3_) [2]. In bromolite chillers, the working medium is water, so their use is limited to producing a refrigerant of about 5 °C. They find their use in air-conditioning systems mainly because of their lower investment than ammonia systems. Ammonia chillers, on the other hand, are mainly used in industrial systems, where temperatures of the produced refrigerant below 0 °C are required. When using ammonia chillers deep freezing to temperatures as low as −60 °C is possible.

The choice of the cooling system is determined by the final economic effect. The use of compressor cooling systems is cheaper than absorption systems. However, the use of absorption chillers can have more favorable operating effects. It is estimated that their primary energy-saving potential is between 30% and 60%. Unfortunately, these results are not often achieved by systems already in operation. Although compressor chillers have a higher cooling capacity (coefficient of performance, COP = 2–5, while for absorption chillers COP = 0.6–1.2), they consume more electricity, which gives an advantage to absorption units. An additional advantage of absorption chillers is the longer service life of absorption units compared to refrigeration units due to the presence of a small number of moving parts in the unit, which also makes servicing much easier.

However, the many advantages of absorption refrigeration do not mean that it is a technology without drawbacks. The corrosive properties, explosiveness, crystallization, and toxicity [3,4,5] of the systems used to date (absorbent + coolant) make it necessary to search for alternative solutions. The search for new, more favorable working pairs has gained importance and is of interest to many research laboratories. In the world literature, several systems are proposed for use in the subject under consideration [6,7,8,9,10,11].

Ionic liquids are a very interesting class of compounds that exhibit unique properties, i.e., extremely low volatility, high heat capacity, non-flammability, high thermal and chemical stability, and total or partial solubility in water and organic solvents [12]. The undoubted advantage of ionic liquid chemistry is the ability to control and refine their properties through the appropriate selection of the cation and anion structure. This allows the design of new target-oriented media. Their low volatility argues for their use as alternative solvents in many chemical industries, including but not limited to chemical synthesis, in the field of extraction and separation, as electrolytes in batteries, lubricants, bactericidal and fungicidal compounds, drug carriers, wood preservatives, or in industrial-scale separation techniques [13,14,15,16].

Many ILs systems as the working fluids in absorption refrigeration are given in the literature [17,18,19,20,21,22,23,24,25,26,27,28,29,30,31,32,33,34]. In this work, we propose three ethanolic solutions of the dimethyl phosphate-based iLs with the following cation: 1-ethyl-1-methylmorpholinium, [C_1_C_2_MOR]^+^, 1-ethyl-1-methylpiperidinium, [C_1_C_2_PIP]^+^, and *N*,*N*,*N*-trimethyl-*N*-ethylammonium, [N_1,2,2,2_]^+^ for future use in this area. The use of ethanol instead of water as a refrigerant will further expand the possibility of using such systems in the temperature range below 273 K. To determine the potential application of iLs and their systems as a working fluid in absorption refrigeration technology it is necessary to outline and describe their basic thermodynamic and physicochemical properties. The characteristics of the systems should include, in particular, measurements of (vapor + liquid) phase equilibria, enthalpy of mixing, heat capacity, and transport properties. Up to now, there are only a few publications concerning the properties of pure dimethyl phosphate-based IL or their binary solution with ethanol. The published work on the VLE data concerns the ethanolic solution of the 1,3-dimethylimidazolium dimethyl phosphate, [C_1_C_1_IM][DMP] [35,36,37], 1-ethyl-3-methylimidazolium dimethyl phosphate, [C_1_C_2_IM][DMP] [38,39], triethylmethylammonium dimethyl phosphate [N_1,2,2,2_][DMP] [40], 1-ethyl-1-methylpyrrolidinium dimethyl phosphate, [C_1_C_2_PYR][DMP] [31] and 1-hydroxy-ethyl-1-methylpyrrolidinium dimethyl phosphate, [C_1_C_2OH_PYR][DMP] [31]. Very few publications deal with measurements of enthalpy of mixing in ionic liquid systems with ethanol; data are available for the following ionic liquids: 1,3-dimethylimizadolium dimethyl phosphate, [C_1_C_1_IM][DMP] [35], 1-ethyl-3-methylimidazolium dimethyl phosphate, [C_1_C_2_IM][DMP] [38], and 1-ethyl-1-methylpiperidinium dimethyl phosphate, [C_1_C_2_PIP][DMP] [41] 1-ethyl-1-methylpyrrolidinium dimethyl phosphate, [C_1_C_2_PYR][DMP] [41], *N*,*N*,*N*-triethyl-*N*-methylammonium dimethyl phosphate, [N_1,2,2,2_][DMP] [41], and 1-hydroxyethyl-1-methyl-pyrrolidinium dimethyl phosphate, [C_1_C_2OH_PYR][DMP] [41]. More literature data concern the measurements of density and viscosity of pure iLs with DMP anion [42,43,44,45,46,47,48,49,50,51,52,53,54] but only in a few of them, there are data on the density or viscosity of an ethanolic solution of iLs [35,38,50,54]. To the best of our knowledge, among the ionic liquids analyzed in this work, there is no literature data for comparison. Therefore, the data presented here and its comparison for other dimethyl phosphate-based IL with the available literature data can be a valuable complement to the state of the art in the area undertaken.

## 2. Results

### 2.1. DSC Measurements

The thermophysical properties of pure iLs were determined at a temperature range from *T* = (173.15 to 373.15) K with a 5 K·min^−1^ heating rate. The experimental data are collected in Table 1 and graphically presented in Figure 1. The list of available thermophysical data of the family of dimethyl phosphate-based iLs is presented in Table 1. The data shows that glass transition temperature was determined for all ion liquids, and the value of *T*_g_ increases in the following series: [C_1_C_2_PYR][DMP] (*T*_g_ = 189.7 K; Δ_g_*C*_p_ = 119.6 J·mol^−1^·K^−1^) < [N_1,2,2,2_][DMP] (*T*_g_ = 194.4 K; Δ_g_*C*_p_ = 154.4 J·mol^−1^·K^−1^) < [C_1_C_2OH_PYR][DMP] (*T*_g_ = 194.9 K; Δ_g_*C*_p_ = 148.1 J·mol^−1^·K^−1^) < [C_1_C_2_PIP][DMP] (*T*_g_ = 211.0 K; Δ_g_*C*_p_ = 103.9 J·mol^−1^·K^−1^) < [C_1_C_2_MOR][DMP] (*T*_g_ = 222.2 K; Δ_g_*C*_p_ = 132.7 J·mol^−1^·K^−1^). The melting temperature was determined for two of the presented iLs. Among dimethyl phosphate-based iLs, the lowest melting point equal to *T*_m_ = 285.2 K (Δ_m_*H* = 0.73 kJ·mol^−1^) was determined for [C_1_C_2_PYR][DMP]. Ionic liquids: [C_1_C_2_PIP][DMP] and [N_1,2,2,2_][DMP] exhibit a similar melting point of about 295 K, whereas, in the case of ammonium-based IL, melting enthalpy is almost seven times higher than for [C_1_C_2_PIP][DMP]. Additionally, a (solid-solid) phase transition temperature of 275.5 K was determined only for [C_1_C_2_PYR][DMP]. According to our best knowledge, the thermophysical properties of pure IL with dimethyl phosphate anion tested in this research have not been reported in the literature; thus, in Table 1, there are no comparisons between the experimental and literature data. The list of available thermophysical data of the family of dimethyl phosphate-based iLs is presented in Table 1.

### 2.2. (Vapor-Liquid) Phase Equilibria Measurements

The VLE data of the ethanolic solution of iLs: [C_1_C_2_MOR][DMP], [C_1_C_2_PIP][DMP], and [N_1,2,2,2_][DMP] have been determined using ebulliometric method at a temperature from *T* = (328.15 to 348.15) K with an increment of 10 K and pressures up to 90 kPa for different ionic liquid mole fraction. The isothermal experimental *P-T-x* data for the tested binary systems are listed in Table 2, Table 3 and Table 4.

As an example, the experimental VLE data together with the NRTL correlations for {[C_1_C_2_MOR][DMP] (1) + ethanol (2)} system is presented in Figure 2. These results for the other two systems are shown in Appendix A (SM).

The vapor-liquid phase equilibria (VLE) in the ethanolic solution of the tested IL are written by the following equation:(1)$$\varphi_{i}y_{i}P=\gamma_{i}x_{i}P_{i}^{S}$$
where *P* is the pressure of the system; for component *i*: $P_{i}^{S}\;$represents the saturated vapor pressure; $x_{i}\;$and $y_{i}$ are the liquid and vapor molar fractions;$\;\varphi_{i}$ is the fugacity coefficient in the gas phase; $\gamma_{i}$ is the activity coefficient in the liquid phase. Due to the low volatility of ionic liquids, the gas phase is assumed to consist only of solvent (ethanol) for IL systems. Herein, $y_{2}$ is approximately equal to 1 and $\varphi_{2}$ is also approximately equal to 1 because of the relatively low pressure. Thus, Equation (1) can be further reduced to the following:(2)$$P=\gamma_{2}x_{2}P_{2}^{S}$$

Experimental data for each tested system shows a typical relationship that is a decrease in the vapor pressure of the ethanolic solution with an increase in the ionic liquid mole fraction. This has to do with the extremely low vapor pressure of a pure ionic liquid. Additionally, as the temperature increases, the vapor pressure in the studied systems increases.

According to Equation (2), the experimental activity coefficient of ethanol ($\gamma_{2}$) in an IL-containing binary liquid mixture was calculated directly from the VLE data and is also presented in Table 2, Table 3 and Table 4. In each case, the determined values of the ethanol activity coefficient were below unity, indicating that the tested systems show negative deviations from the ideal behavior. This is due to the occurrence of stronger intermolecular interactions between IL and ethanol compared to (ethanol–ethanol) or (IL–IL) interactions. The negative deviation from Raoult’s law is larger for higher IL mole fractions. It is worth mentioning that preferentially, IL used as an absorbent in absorption refrigeration technology should exhibit a powerful ability to absorb refrigerant (ethanol). Therefore, good working pairs are those presenting a highly negative deviation from Raoult’s law. In this work, for each system presented here, the activity coefficients of ethanol (*γ*_2_) are lower than unity; thus, each system presents a negative deviation from the ideal solution, which makes the working fluids proposed in this work potentially promising for applications in the area of interest undertaken.

A comparison of the VLE phase diagrams of {IL (1) + ethanol (2)} system under study at a temperature of 348.15 K within a pressure range up to 90 kPa and the vapor pressure of the ideal solution is shown in Figure 3. Such a comparison makes it possible to determine the effect of the structure of the cation of the ionic liquid on the vapor pressure, consequently, on the value and type of deviation of {IL + ethanol} systems from ideality. It was shown that among ethanolic systems investigated here, the vapor pressure increases in the following series: [N_1,2,2,2_][DMP] < [C_1_C_2_PIP][DMP] < [C_1_C_2_MOR][DMP]. Among ionic liquids presented, the highest negative deviation from Raoult’s law, so the strongest interactions with ethanol shows [N_1,2,2,2_][DMP] and the weakest in {[C_1_C_2_MOR][DMP] (1) + ethanol (2)}. It is likely that in the case of an ammonium-based ionic liquid, intermolecular interactions between the ILs anion and ethanol play a key role. In addition, the short alkyl chains present in the cation of the ionic liquid promote stronger packing of alcohol molecules in the ionic liquid and the occurrence of van der Waals interactions. The weaker intermolecular interactions present in the case of an ionic liquid with a piperidinium cation are probably a consequence of its ring structure. It affects the formation of steric hindrance and weakens the interaction between IL and ethanol. In the case of the ionic liquid [C_1_C_2_MOR][DMP], the system shows the highest vapor pressure, therefore, the lowest deviation from ideality. It is speculated that the presence of an oxygen atom in the cation of the ionic liquid affects the occurrence of stronger interactions between ionic liquid molecules (IL–IL) than between the ionic liquid and the solvent (IL–ethanol).

A broader comparison of the effect of the ILs cation structure for the family of the dimethyl phosphate IL, taking into account the effect of the core structure and the presence of a hydroxyl group in the substituent on the vapor pressure of the {IL + ethanol} system is also shown in Figure 3. It was shown that for the presented family of the DMP-based IL, the vapor pressure increases in the following series: [N_1,2,2,2_][DMP]~[C_1_C_1_IM][DMP]~[C_1_C_2_IM][DMP] < [C_1_C_2_PYR][DMP]~[C_1_C_2_PIP][DMP] < [C_1_C_2_MOR][DMP] < [C_1_C_2OH_PYR][DMP]. The highest negative deviation from Raoult’s law, thus, the strongest (IL-ethanol) interaction was determined in the case of IL with ammonium cation. A similar effect was observed for 1,3-dimethylimidazolium- and 1-ethyl-3-methylimidazolium cations. It was noted that the extension of the alkyl chain by one CH_2_ group in the imidazolium cation of IL has no significant effect on the vapor pressure in the studied system. This observation also holds true when increasing the core size of the cation from pyrrolidinium to piperidinium. Moreover, the vapor pressure in the system with [C_1_C_2OH_PYR][DMP] is significantly higher compared to the aliphatic analog, [C_1_C_2_PYR][DMP]. It is speculated that the presence of a hydroxyl group in an ILs cation creates a greater possibility for a molecule to form hydrogen bonds between the molecules of the ionic liquid, thereby weakening the (IL-ethanol) interaction. Similarly, in the case of an ionic liquid with a morpholinium cation, the presence of an oxygen atom in the structure of the IL cation probably promotes the occurrence of stronger interactions (IL-IL) compared to (IL-ethanol) intermolecular interactions.

In our latest paper [31], it was shown that {[N_1,2,2,2_][DMP] (1) + ethanol (2)} exhibit the highest vapor pressure among dimethyl phosphate-based IL. It was stated at the time that the ammonium cation substituted with alkyl groups is more aliphatic compared to the imidazolium cation, which prevents the formation of intermolecular hydrogen bonds with ethanol molecules. The small number of literature points under the considered temperature conditions (*T* = 348.15 K) prompted the authors of this work to conduct a more extensive study of VLE in this system. The experimental data compared with available literature data, is shown in Figure 3. The obtained results differ slightly from the literature data presented by Shen et. al. [40]. The reason for the reduction in vapor pressure in the {[N_1,2,2,2_][DMP] + ethanol} system presented in this work is the water content of the pure ionic liquid. In the present work, the ionic liquid was synthesized in the laboratory, and the claimed water content is 8000 ppm, while Shen. et al. [40] declare the water content to be 664 ppm. From the literature studies [40] on VLE measurements in {water (1) + ethanol (2) + IL (3)} ternary systems, it is clear that an increase in water content results in a decrease in vapor pressure in the system.

Since the goal of the ongoing research is to search for working fluids alternative to the aqueous solution of lithium bromide, in addition to the vapor pressure data of {IL + ethanol} systems, Figure 3 also shows VLE data for {LiBr + water} system [55]. Although the tested iLs show a high ability to absorb ethanol, the vapor pressure under the same temperature conditions is higher than that of the commercially used system in refrigeration. It is worth considering the possibility of future use of ionic liquids as absorbents in cooling devices since they are designed compounds and exhibit several unique properties, and the commercially used systems are not without drawbacks. Apart from a few literature data for the ethanolic solution of ammonium-based IL, we have not found the literature data on (vapor + liquid) phase equilibria measurements for the systems analyzed in this work.

The NRTL equation for the excess Gibbs energy for the binary system is the following:(3)$$\frac{G^{E}}{RT}=x_{1}x_{2}\left[\frac{\tau_{21}G_{21}}{x_{1}+G_{21}x_{2}}+\frac{\tau_{12}G_{12}}{G_{12}x_{1}+x_{2}}\right]$$
where:(4)$$G_{12}=\exp\left(-\alpha_{12}\tau_{12}\right)\;\;\;\;\;\;\;\;\;\;\;G_{21}=\exp\left(-\alpha_{21}\tau_{21}\right)$$
(5)$$\tau_{12}=\frac{g_{12}-g_{22}}{RT}\;\;\;\;\;\;\;\;\;\;\;\;\;\;\tau_{21}=\frac{g_{21}-g_{11}}{RT}$$
where $g_{12}-g_{22}=\Delta g_{12\;}$ and $g_{21}-g_{11}=\Delta g_{21}$ are the binary interactions parameters. For isothermal data sets, temperature dependence of the parameters is represented as follows:(6)$$\;\Delta g_{12\;}=\Delta g_{12}^{A}T+\Delta g_{12}^{B}\;\;\Delta g_{21\;}=\Delta g_{21}^{A}T+\Delta g_{21}^{B}$$

For a complete *T*-*P*-*x*-*y* data set, a total of the following five parameters that is: $\Delta g_{12}^{A},\;\Delta g_{12}^{B},\;\Delta g_{21}^{A},\Delta g_{21}^{B},\;\alpha_{12}=\alpha_{21}$ are determined. From a developmental point of view, activity coefficients,$\;\gamma_{i}$ derived from excess Gibbs energies but in practice and in this work, the process is reversed and $G^{E}$ is tested from knowledge of activity coefficient.
(7)$$\ln\gamma_{2}=x_{1}^{2}\left[\tau_{12}{\left(\frac{G_{12}}{x_{2}+x_{1}G_{12}}\right)}^{2}+\frac{\tau_{21}G_{21}}{{\left(x_{1}+x_{2}G_{21}\right)}^{2}}\right]$$

The NRTL parameters were calculated by minimalized function given by the following:(8)$$F=\overset{n}{\underset{i=1}{\sum }}{\left(ln\gamma_{2}^{calc}-ln\gamma_{2}^{exp}\right)}^{2}$$

The adjustable parameters of the following equation: $\Delta g_{12}^{A},\Delta g_{12}^{B},\Delta g_{21}^{A},\Delta g_{21}^{B},\;\alpha_{12}=\alpha_{12}$ along with the average absolute deviation (AAD) given by the following equation:(9)$$\mathrm{AAD}=\frac{100}{n}\overset{n}{\underset{i=1}{\sum }}\left|\frac{P^{exp}-P^{cal}}{P^{exp}}\right|$$
are listed in Table 5.

In graphical form, the experimental *P-T-x* data for {IL (1) + ethanol (2)} system together with the correlation results using the NRTL model are shown in Figure 2 and Figure 3.

### 2.3. Density and Dynamic Viscosity Data

The liquid density and dynamic viscosity data of pure iLs: [C_1_C_2_MOR][DMP], or [C_1_C_2_PIP][DMP], or [N_1,2,2,2_][DMP] and its ethanolic solution were investigated in a whole concentration range at a temperature within (293.15 and 338.15) K with an increment of 5 K. The experiment was performed under ambient pressure, and the experimental data are collected in Table 6, Table 7 and Table 8.

Temperature dependence of the liquid density for each system under study was described using the following equation:(10)$$\rho =\rho_{0}\exp\left[-\alpha_{p}\left(T-T_{0}\right)\right],\;T_{0}=298.15\;K$$

The root mean square error (RMSE) is expressed by
(11)$$\mathrm{RMSE}=\;\sqrt{\frac{\sum_{i=1}^{N}{\left(\rho^{calc}-\rho^{exp}\right)}^{2}}{N-P}}$$

The value of the parameters *ρ*_0_ and *α*_p_ along with RMSE are collected in Table 9.

The maximum RMSE value between the experimental density data and calculated values was determined to be 0.0003 for each binary system under study, which indicates excellent compatibility between these data. Experimental liquid density data versus temperature and composition for {[C_1_C_2_MOR][DMP] (1) + ethanol (2)} is presented in Figure 4 as an example. The solid lines presented in Figure 4a correspond to the results of calculations using Equation (10). The results for the other two systems are shown graphically in Appendix A.

The composition dependence of the liquid density for each system under study was described using a fourth-degree polynomial with the linear temperature dependence of the parameters of the polynomial.
(12)$$\rho /\left(g\cdot {\mathrm{cm}}^{-3}\right)=\left(aT/K+a^{\prime }\right)x^{4}+\left(bT/K+b^{\prime }\right)x^{3}+\left(cT/K+c^{\prime }\right)x^{2}+\left(dT/K+d^{\prime }\right)x+\left(eT/K+e^{\prime }\right)$$

The root mean square error (RMSE) is expressed by the following:(13)$$RMSE=\sqrt{\frac{\sum_{i=1}^{N}{\left(\rho^{calc}-\rho^{exp}\right)}^{2}}{N-P}}$$

The parameters of the polynomial described by Equation (12), together with the standard deviation, are collected in Table 10. The maximum *σ* value between the experimental and calculated density data was determined to be 0.03% for each binary system under study, which indicates excellent compatibility between these data. Experimental liquid density data versus composition for {[C_1_C_2_MOR][DMP] (1) + ethanol (2)} is presented in Figure 4b as an example. The solid lines presented in Figure 4b correspond to the results of calculations using Equations (12) and (13). The results for the other two systems are shown graphically in Appendix A.

Experimental density and viscosity data show that both pure iLs and their ethanolic solutions exhibit higher liquid density than ethanol, and, as expected, density and viscosity decrease with increasing temperature. This behavior is related to thermal expansion; the fluid density decreases and the intermolecular interactions become weaker due to the increase in the mutual distances between the molecules, and, therefore, the viscosity also decreases. Additionally, the addition of ethanol leads to a decrease in both mixture density and viscosity at each temperature.

Density and viscosity are fundamental physical properties of the solvent and tested medium and are extremely important in the design and optimization of any application, including absorption refrigeration technology, since they may be correlated with the medium fluidity and mass transfer. For this reason, both experimental studies and a discussion of values based on available literature data are warranted. We found no literature data for binary solution composed of ethanol and dimethyl phosphate-based iLs as well as the density of pure ionic liquids under study. Based on experimental and literature data for other dimethyl phosphate-based iLs, it was possible to discuss the effect of the cation structure on the liquid density of the ethanolic solution. The comparison of the experimental liquid density data for {[cation][DMP] (1) + ethanol (2)} system with the literature for other dimethyl phosphate-based ILs solutions at temperature *T* = 298.15 K is presented in Figure 5.

The comparison shows the desirable characteristics of the studied systems from the point of view of future use as working fluids in absorption refrigeration technology because this liquid density value is lower than those for lithium bromide aqueous solution, commercially used as a working fluid in this area. It was shown that the liquid density of {IL (1) + ethanol (2)} decreases in the following series: [C_1_C_1_IM][DMP] > [C_1_C_2_MOR][DMP] > [C_1_C_2OH_PYR] [DMP] > [C_1_C_2_IM] [DMP] > [C_1_C_2_PYR][DMP] > [C_1_C_2_PIP][DMP]~[C_1_C_4_IM][DMP] > [N_1,2,2,2_][DMP]. The increase in the length of the alkyl chain in the imidazolium ring causes a decrease in density. The same trend is observed when the cyclic chain is increased from a five-member pyrrolidinium ring to a six-member piperidine ring. The presence of a hydroxyl group in the pyrrolidinium cation results in a significant increase in density. In addition, the presence of an oxygen atom in the morpholinium cation of the ionic liquid increases the density of the binary system with ethanol compared to the system with piperidinium-based IL.

From Figure 5, the ethanolic system with the ammonium-based ionic liquid shows the lowest density. These data, supplemented by the promising vapor pressure values presented earlier, allow us to conclude that the {[N_1,2,2,2_][DMP] + ethanol} system is the most promising for application in the area being undertaken.

Based on the liquid density data for pure compounds and the binary systems, the excess molar volumes ($V^{E}$) for {IL (1) + ethanol (2)} solutions under study were calculated at each temperature. Obtained data are given in Table 6, Table 7 and Table 8. The temperature and composition dependence on the $V^{E}$ for each system under study is graphically presented in Figure 6.

For all systems under work, the value of $V^{E}$ is negative in the entire composition range at each temperature, which indicates the occurrence of stronger (IL–ethanol) compared to (ethanol–ethanol) or (IL–IL) interactions. The $V^{E}$ values can be explained by many contributions, such as contraction because of specific interactions between IL and ethanol or differences in molecule sizes.

The $V^{E}$ values decrease with an increase in temperature. The minimum of the excess molar volumes varies from −0.9800 cm^3^·mol^−1^ (*x*_1_ = 0.2005) at *T* = 293.15 K to −1.5163 cm^3^·mol^−1^ (*x*_1_ = 0.1992) at *T* = 338.15 K for {[C_1_C_2_MOR][DMP] (1) + ethanol (2)}, from −0.9233 cm^3^·mol^−1^ (*x*_1_ = 0.2000) at *T* = 293.15 K to −1.4504 cm^3^·mol^−1^ (*x*_1_ = 0.2000) at *T* = 338.15 K for {[C_1_C_2_PIP] [DMP] (1) + ethanol (2)}, and from -0.7911 cm^3^·mol^−1^ (*x*_1_ = 0.1999) at *T* = 293.15 K to −1.2597 cm^3^·mol^−1^ (*x*_1_ = 0.1999) at *T* = 338.15 K in the case of [N_1,2,2,2_][DMP] ethanolic solution.

A comparison of *V*^E^ values at 298.15 K for the three ethanol solutions presented in this work is shown graphically in Figure 6d. It shows that below the IL mole fraction *x*_1_ = 0.5, the *V*^E^ values increase in the following series: [C_1_C_2_MOR][DMP] < [C_1_C_2_PIP][DMP] < [N_1,2,2,2_][DMP] while for compositions above *x*_1_ = 0.5 the increase in *V*^E^ is observed in the following order: [C_1_C_2_PIP] [DMP] < [N_1,2,2,2_][DMP] < [C_1_C_2_MOR][DMP].

The values of the excess molar volume were correlated with the Redlich–Kister equation with the temperature dependence of the parameters as follows:(14)$$V^{E}/({\mathrm{cm}}^{3}\cdot {\mathrm{mol}}^{-1})=x_{1}x_{2}\overset{k}{\underset{k=0}{\sum }}A_{k}{\left(x_{1}-x_{2}\right)}^{k}=x_{1}x_{2}\left[A_{0}+A_{1}\left(x_{1}-x_{2}\right)+A_{2}{\left(x_{1}-x_{2}\right)}^{2}\right]$$
where *x*_1_ and *x*_2_ is the ionic liquid and ethanol mole fraction, respectively; $V^{E}$/(cm^3^·mol^−1^) is the excess molar volume; $A_{k}$ is the temperature dependence parameters given as following:(15)$$A_{k}=a_{k}+b_{k}T$$

The root mean square error (RMSE) is expressed by the following:(16)$$RMSE=\sqrt{\frac{\sum_{i=1}^{n}{\left(V_{i}^{E\left(\exp\right)}-V_{i}^{E\left(calc\right)}\right)}^{2}}{\left(n-p\right)}}$$
where *n* is the number of experimental points and *p* is the number of coefficients.

The value of the Redlich–Kister parameters, $A_{k}$, along with the standard deviation, are given in Table 11.

The combined standard uncertainty of excess volume of mixing (*V*^E^) was determined using the error propagation law, using Equations (17) and (18) was calculated to be 0.008 cm^3^·mol^−1^.
(17)$$u_{c}\left(V^{E}\right)=\sqrt{{\left(\left(\frac{\partial V^{E}}{\partial x_{1}}\right)\Delta x_{1}\right)}^{2}+{\left(\left(\frac{\partial V^{E}}{\partial M_{1}}\right)\Delta M_{1}\right)}^{2}+{\left(\left(\frac{\partial V^{E}}{\partial M_{2}}\right)\Delta M_{2}\right)}^{2}+{\left(\left(\frac{\partial V^{E}}{\partial \rho }\right)\Delta \rho \right)}^{2}+{\left(\left(\frac{\partial V^{E}}{\partial \rho_{1}}\right)\Delta \rho_{1}\right)}^{2}+{\left(\left(\frac{\partial V^{E}}{\partial \rho_{2}}\right)\Delta \rho_{2}\right)}^{2}}$$
and
(18)$$u_{c}\left(x\right)=\sqrt{{\left(\left(\frac{\partial x}{\partial m_{1}}\right)\Delta m_{1}\right)}^{2}+{\left(\left(\frac{\partial x}{\partial m_{2}}\right)\Delta m_{2}\right)}^{2}}$$
where $M_{1}$, $M_{2}$ are the molar masses of IL and ethanol, respectively; $\Delta M_{1}$, $\Delta M_{2}$ are the error of molar masses of IL and ethanol, respectively ($\Delta M_{1}=0$, $\Delta M_{2}=0)$; $\rho_{1}$, $\rho_{2}$ are the densities of IL and ethanol, respectively; $x_{1}$, $x_{2}$ is the composition of IL and ethanol, respectively; $\Delta x_{1}$, $\Delta x_{2}$ is the uncertainty in composition determination of the mole fraction of IL and ethanol, respectively, $m_{IL}$, $m_{s}$ is the mass of ionic liquid and solvent, respectively; $\Delta m_{IL}$, $\Delta m_{s}$ is the weighing error (Δm = 0.0001 g); Δρ is the uncertainty of density measurement.

Dynamic viscosity data of {[C_1_C_2_MOR][DMP], or [C_1_C_2_PIP][DMP], or [N_1,2,2,2_] [DMP] (1) + ethanol (2)} binary systems was measured at a whole composition range at temperatures from *T* = (293.15 to 338.15) K with an increment of 5 K. The experimental data are collected in Table 6, Table 7 and Table 8. The temperature and composition dependence on dynamic viscosity data for {[C_1_C_2_MOR][DMP] (1) + ethanol (2)} are graphically presented in Figure 7 as an example. The other two ethanolic solutions under study are shown in Appendix A.

As expected, for each system under study, dynamic viscosity data decreases with an increasing temperature and increasing ethanol content. The comparison of dynamic viscosity data for an ethanolic solution under study at a temperature of 298.15 K is presented in Figure 8. It can be observed that the viscosity values for the tested systems decrease in the following series: [C_1_C_2_MOR][DMP] > [C_1_C_2_PIP] [DMP] > [N_1,2,2,2_][DMP], therefore, the most favorable results from the point of view of technological application in as working fluids were obtained for the ethanol mixture of ammonium-based ionic liquid. Additionally, among the ethanolic solution of dimethyl phosphate-based iLs, dynamic viscosity decreases in the following series: [C_1_C_2_MOR][DMP] > [C_1_C_2_PIP][DMP] > [C_1_C_2OH_PYR][DMP] > [C_1_C_4_IM][DMP] > [N_1,2,2,2_][DMP] > [C_1_C_1_IM][DMP] > [C_1_C_2_IM][DMP] > [C_1_C_2_PYR] [DMP].

Within investigated temperature range, the dynamic viscosities for pure iLs vary from *η* = 17,870 mPa·s at *T* = 293.15 K to 289.6 at *T* = 338.15 K for [C_1_C_2_MOR] [DMP], from 3276 mPa·s at *T* = 293.15 K to 136.7 mPa·s at *T* = 338.15 K for [C_1_C_2_PIP][DMP] and from 679.6 mPa·s at *T* = 293.15 K to 41.46 mPa·s at *T* = 338.15 K in the case of [N_1,2,2,2_][DMP]. The comparison of the dynamic viscosity value for pure dimethyl phosphate-based IL at a temperature of 298.15 K is graphically presented in Figure 9. Experiment shows that the dynamic viscosity value for pure iLs decreases in the following series: [C_1_C_2_MOR][DMP] > [C_1_C_2_PIP][DMP] > [C_1_C_2OH_PYR][DMP] > [N_1,2,2,2_][DMP] > [C_1_C_2_PYR][DMP]. The highest viscosity equal to 10,070 mPa·s was determined for [C_1_C_2_MOR][DMP], and it is almost 60 times higher than that for [C_1_C_2_PYR][DMP].

Temperature dependence of the dynamic viscosity for the binary systems under study was described using the following Andrade-type equation [56]:(19)$$\ln\eta =A-\frac{B}{T}$$

Parameter B is described by $B=\frac{-E_{a}}{R}$ where *R* is the gas constant and *E*_a_ is the flow activation energy.

Correlation parameters were calculated based on the relative residuals and the parameters were calculated by minimalized function given by the following:(20)$$F=\overset{n}{\underset{i=1}{\sum }}{\left(\frac{\eta^{calc}-\eta^{exp}}{\eta^{exp}}\right)}^{2}$$

The root mean square deviation (RMSE) is expressed as following:(21)$$\mathrm{RMSE}=\sqrt{\frac{\sum_{i=1}^{N}{\left(\eta^{calc}-\eta^{exp}\right)}^{2}}{N-P}}$$

The values of parameters *A* and *B*, along with RMSE, are given in Table 12. The calculation is graphically presented as solid lines in Figure 7.

## 3. Discussion

Thermodynamic and physicochemical properties of three pure ionic liquids and their ethanolic solution versus temperature and composition were presented in this study. Binary systems composed of IL and ethanol were analyzed for possible future use as working fluids in absorption refrigeration technology. In the studied systems, the ionic liquid would act as an absorbent and ethanol as a circulating agent. This is to search for alternative systems to the commercially used lithium bromide aqueous solution. An additional benefit would be the ability to operate the device at temperatures below 273.2 K due to the use of ethanol instead of water as the circulating medium. In the design of the new working fluid, it is desirable that the mixture exhibits negative deviations from Raoult’s law (resulting in easier refrigerant absorption) and has as low density and viscosity values as possible, which allows easier mass and heat transfer. In this work, three binary solutions composed of [C_1_C_2_MOR][DMP], or [C_1_C_2_PIP][DMP], or [N_1,2,2,2_][DMP] and ethanol were considered as an alternative working pair for the absorption refrigeration. Using the DSC technique, the thermophysical properties of pure iLs were determined. The experimental value was compared to those for other dimethyl phosphate-based iLs. For each IL under study, glass temperature and the heat capacity at the glass transition temperature were determined. It was shown that [C_1_C_2_PIP][DMP] and [N_1,2,2,2_][DMP] exhibit similar melting temperatures, whereas the melting enthalpy for IL with ammonium cation is almost 7-times higher than for [C_1_C_2_PIP][DMP].

Thermodynamic and physicochemical properties, including vapor pressure, liquid density, and dynamic viscosity of binary systems composed of IL and ethanol, were experimentally determined as a function of temperature and composition. The experimental data was successfully correlated using the appropriate equations. Each system exhibits negative deviations from the ideal solution. The highest negative deviation from Raoult’s equation was determined for the ethanolic solution of [N_1,2,2,2_][DMP]. The comparison of VLE data investigated here with those for other dimethyl phosphate-based iLs shows that the vapor pressure increases in the following series: [N_1,2,2,2_][DMP]~[C_1_C_1_IM][DMP]~[C_1_C_2_IM] [DMP] < [C_1_C_2_PYR][DMP]~[C_1_C_2_PIP][DMP] < [C_1_C_2_MOR][DMP] < [C_1_C_2OH_PYR][DMP]. Despite the characteristics desirable for applications in the area being undertaken, the vapor pressures of the proposed systems are higher compared to the commercially used aqueous lithium bromide solution. The correlation of the experimental data with the NRTL equation resulted in good agreement between experimental and calculated values.

Both liquid density and dynamic viscosity decrease as temperature and ethanol content in the systems increase. Experimental density and dynamic viscosity data for the ethanolic solution presented here were compared to those for other dimethyl phosphate-based iLs. It was shown that the density values decrease in the following series: [C_1_C_1_IM][DMP] > [C_1_C_2_MOR][DMP] > [C_1_C_2OH_PYR][DMP] > [C_1_C_2_IM][DMP] > [C_1_C_2_PYR][DMP] > [C_1_C_2_PIP][DMP]~[C_1_C_4_IM][DMP] > [N_1,2,2,2_][DMP], while viscosity decreases as follows: [C_1_C_2_MOR][DMP] > [C_1_C_2_PIP][DMP] > [C_1_C_2OH_PYR][DMP] > [C_1_C_4_IM] [DMP] > [N_1,2,2,2_][DMP] > [C_1_C_1_IM][DMP] > [C_1_C_2_IM][DMP] > [C_1_C_2_PYR][DMP]. Based on liquid density data, the excess molar volumes were calculated and correlated using the Redlich–Kister equation with the temperature dependence of parameters. Since the main objective of the work is to search for working fluids for future use as an alternative to {LiBr + water} refrigeration systems, the experimental data presented were compared with those for an aqueous solution of lithium bromide, conventionally used in industrial-scale refrigeration technologies. The comparison shows that liquid density data for each binary system is lower than those for {LiBr + water} (see Figure 5). Dynamic viscosity for aqueous lithium bromide solution was determined within a narrow range of composition showing a steep course, and at higher concentrations of lithium bromide, the values would be diametrically higher than for the systems proposed in this work (see Figure 8). Resuming, the ethanol solutions of ionic liquid analyzed in this work, especially [N_1,2,2,2_][DMP], exhibit a high potential for use as a working fluid in absorption cooling technology.

## 4. Materials and Methods

### 4.1. Materials

The ethanolic solutions of the dimethyl phosphate-based iLs, namely, 1-ethyl-1-methylmorpholinium dimethyl phosphate (abbreviated as: [C_1_C_2_MOR][DMP]), 1-ethyl-1-methylpiperidinium dimethyl phosphate ([C_1_C_2_PIP][DMP]) and *N*,*N*,*N*-triethyl-*N*-methylammonium dimethyl phosphate ([N_1,2,2,2_][DMP]) were investigated. The sample of [C_1_C_2_PIP][DMP] with an initial mass fraction purity 0.970 was purchased from IoLiTec. The synthesis procedure of *N*,*N*,*N*-triethyl-*N*-methylammonium dimethyl phosphate was presented in our latest work [41]. Here, 1-ethyl-1-methylmorpholinium dimethyl phosphate was synthesized. The structure of the final product was analyzed by ^1^H NMR and ^13^C NMR spectra. A detailed description of the synthesis procedure and NMR analysis is given below.

#### Synthesis of 1-ethyl-1-methylmorpholinium dimethyl phosphate, [C_1_C_2_MOR][DMP]

A mass of 115.153 g (1.000 mol) 4-ethylmorpholine (Alfa Aesar, CAS: 100-74-3, purity: 98%) and 121.118 g (0.865 mol) trimethyl phosphate (ACROS Organics, CAS: 512-56-1, purity 99%) was added directly to a 500 mL flask. The synthesis was carried out under reflux at 398.2 K for 24 h under a nitrogen atmosphere. The mixture was extracted several times with ethyl acetate. The residual ethyl acetate was evaporated on the rotary evaporator, and the product was dried in a vacuum oven at temperature *T* = 353.2 K for 24 h. A mass of 199.143 g of dark brown, viscous liquid was obtained. The yield of the synthesis was 90.33%.

^1^H NMR (500 MHz, D_2_O, 298 K): δ (ppm): Cation: 4.07–4.04 (4H, m, H_2_), 3.56 (2H, q, J_CH2,CH3_ = 7.5 H, CH_2_), 3.53–3.44 (4H, m, H_3_), 3.17 (3H, s, N-CH_3_), 1.39 (3H, tt, C–CH_3_); Anion: 3.59 (6H, d, JH, P = 10.5 Hz, CH_3_).

^13^C NMR (125 MHz, CD_3_CN, 298 K): δ (ppm): Cation: 61.27 (C_2_), 61.27 (CH_2_), 60.03 (C_3_), 46.91 (N–CH_3_), 7.60 (C–CH_3_); Anion: 52.23 and 52.18 (CH_3_).

The NMR spectra are presented in Appendix A. The structures of the iLs tested in this work are presented in Table 13. To remove remaining volatile chemicals and to decrease water content before measurements, every IL was drained for 48 h in a vacuum drying oven (Binder, model VD 23) at a temperature *T* = 373 K and under reduced pressure (*P* = 4∙10^−4^ mbar) obtained by a vacuum pump (Vacuubrand RZ 6). The water mass fraction of the dried IL was determined using Karl-Fischer titration (model SCHOTT Instruments TitroLine KF). The sample description, including the purities, water content, and purification methods, is presented in Table 14.

Binary systems were prepared by weighing the pure components on a Mettler Toledo AB 204-S balance, with a precision of 1∙10^−4^ g. Due to the high water absorption of ionic liquids, to prevent variations in the composition, the solutions were weighed just before the measurements.

### 4.2. Experimental Apparatus and Procedure

#### 4.2.1. Differential Scanning Calorimetry (DSC)

Thermophysical properties of pure iLs, including temperature (*T*_m_) and enthalpy of melting (Δ_m_*H*), glass transition temperature (*T*_g_), and heat capacity at the glass transition temperature (*Δ*_g_*C*_p_), as well as temperature (*T*_tr_) and enthalpy (Δ_tr_*H*) of (solid + solid) phase transition, were determined by DSC 1 STAR^e^ System (Mettler Toledo) calorimeter, which used differential scanning calorimetry (DSC) technique. The calibration of the apparatus was carried out by measuring the samples of 99.9999 mol% purity indium and high-mass-purity *n*-heptane (Sigma-Aldrich (St. Louis, MO, USA), ≥99.5%), *n*-octane (Sigma-Aldrich, ≥99%), *n*-decane (Sigma-Aldrich, ≥99%), ethylbenzene (Sigma-Aldrich, ≥99%), cyclohexane (POCH, ≥99.5%), *n*-dodecane (Acros Organics, ≥99%), *n*-octadecane (Aldrich, ≥99%), naphthalene (Acros Organics, ≥99%), and water (Millipore, κ < 0.05 mS·cm^−1^). Calibration measurements were made at the heating/cooling rate of 5 K·min^−1^ in a temperature range from *T* = (180 to 430) K. The uncertainties are as follows: *u*(*T*) = 0.3 K, *u*(*ΔH*) = 3.3 J∙g^−1^. Measurements of all examined IL samples were performed in heating mode at the heating rate of 5 K·min^−1^. A sample of the tested ionic liquid weighing about 10 mg was placed in an aluminum pan and an empty hermetic aluminum pan was used as a reference. The apparatus includes a liquid nitrogen cooling system, and the measurements were made in an inert atmosphere with a nitrogen flux of about 200 mL/min. The experiments were carried out within the temperature range from (173.15 to 373.15) K. The experimental data were analyzed using STARe DB V12.00 software (Bhopal, India).

#### 4.2.2. (Vapor-Liquid) Phase Equilibria Measurements

The isothermal vapor-liquid phase equilibrium (VLE) measurements were performed using an ebulliometric method, where the main part of the apparatus is a specially designed ebulliometer [57]. The mixture of ionic liquid and ethanol placed in the device is in continuous movement forced by the operation of the Cottrell pump. The superheated two-phase mixture gets decompressed, and it is thrown through the tube onto the socket of the thermometer. It is assumed that the expansion in the equilibrium chamber causes the loss of excess heat of the superheated liquid, which is used to evaporate the additional amount of liquid. As a result, the equilibrium temperature is established on the walls of the thermometric socket under the conditions of pressure in the system. By changing the pressure in the system, one can set the temperature of the measurement. Measurements were made at the following three constant temperatures: 348.15, 338.15, and 328.15 K. In the equilibrium chamber, the two-phase mixture is separated into the following two streams: a steam flow and a liquid flux. Ionic liquids have a negligibly low vapor pressure, so the steam flow consists only of ethanol. Before the two streams are combined, samples can be taken from the gas phase and liquid phase reservoirs. Earlier, the steam flow enters the condenser. A very important role is played by thorough mixing of the combined streams. For this purpose, glass tubes were designed in an appropriate way and stabilizing system comprising the isolated container with a volume of 50 dm^3^ enabled the pressure to be kept constant within 0.1 kPa and to dampen the pressure fluctuations caused by the bumping of the liquid boiling in the ebulliometer or by the variation of the temperature of the surroundings was implemented. The equilibrium temperature was measured with a resistance thermometer (type P-550, Roth, Germany) with a precision of 0.01 K. The pressure was measured with the precision of 0.1 kPa by a tensiometric vacuum meter (type CL 300, ZEPWN, Poland). The thermometer and the manometer were calibrated by measuring the boiling points as a function of pressure for *n*-octane, ethanol, and water and compared to values obtained from the literature [58]. The uncertainty was at the level of *u*(*p) =* 0.2 kPa and *u*(*T*) = 0.05 K. The composition of samples taken from the ebulliometer was determined densimetrically based on previous density measurements in two-component systems using the Anton Paar GmbH 4500 vibrating-tube densimeter (Graz, Austria) with an accuracy of 1·10^−5^ g·cm^−3^ at each temperature. Only the liquid phase was sampled because, as previously mentioned, the gas phase consisted only of pure ethanol. A calibration curve of density vs. mole fraction of IL was made and the uncertainty in the mole fraction composition was better than 2·10^−3^. The VLE measurement error was caused by an error related to incorrect indications of the thermometer, error related to incorrect indications of the manometer, error resulting from the calibration curve of the manometer, error related to the accuracy of determining the mole fraction of the liquid phase, error related to other factors influencing the determination of the equilibrium temperature, such as the following: imperfection of mixing inside the measuring cell, which creates a concentration gradient, disturbance of the equilibrium conditions during sampling, or impurities present in the sample. The total measurement error is the sum of the factors mentioned above. The combined uncertainty of the method used for the VLE estimation is larger than the instrument error and was estimated at 0.5 kPa.

#### 4.2.3. Density Measurement

The liquid density of pure ILs and their ethanolic solutions were determined at a temperature range from *T* = (293.15 to 338.15) K with an increment of 5 K at ambient pressure.

The densities were measured on an Anton Paar GmbH 4500 densimeter using vibrating tube method. Measurements were made at different temperatures with a built-in thermostat. The precision in the temperature control (internally of 0.01 K) is provided by two integrated Pt 100 platinum thermometers. The apparatus was calibrated at atmospheric pressure using double-distilled degassed water and air. The densimeter is precise to within 1·10^−5^ g·cm^−3^, and because of the purity of IL, estimated standard uncertainty of the density measurement, *u(ρ)* is better than 5·10^−3^ g·cm^−3^.

#### 4.2.4. Dynamic Viscosity Measurements

Viscosity measurements for pure ILs and their ethanolic solution were carried out using DVNext Wells-Brookfield Cone Rheometer (Middleborough, United States of America). The experiment was performed within the temperature range from *T* = (293.15 to 338.15) K with an increment of 5 K. The apparatus consists of torque measuring system, where a calibrated beryllium-copper spring is the main part. The spring is connected to the drive mechanism of a rotating cone, and it senses the resistance to rotation caused by the presence of a liquid sample between the cone and a stationary flat plate. The resistance to the rotation of the cone produces a torque that is proportional to the shear stress in the fluid. Then, the reading is converted to centipoise units from pre-calculated range charts. The tested pure ionic liquids and their solutions with ethyl alcohol are Newtonian liquids, which means that viscosity does not depend on the shear rate. It was checked before every measurement. The system was previously calibrated to the standards by the manufacturer, and calibration was verified using appropriate standards. The mean standard deviation of viscosity was determined based on standard measurements, and the relative uncertainty, *u*_r_*(η)* value, was determined to be 0.03. The apparatus is accurate to within 1.0% of the full-scale range reproducibility is within 0.2%. The working temperature range is from 0 °C to 100 °C. Two cones were used for viscosity measurements. One for measurements above 250 mPa·s with a cone radius of 0.012 m and a cone angle of 1.5°, and the other for measurements below 250 mPa·s, with a radius of 0.024 m and a cone angle of 0.8°. The gap between the cone truncation and the plate was 12.7·10^–6^ m.

## Figures and Tables

**Figure 1 molecules-28-01940-f001:**
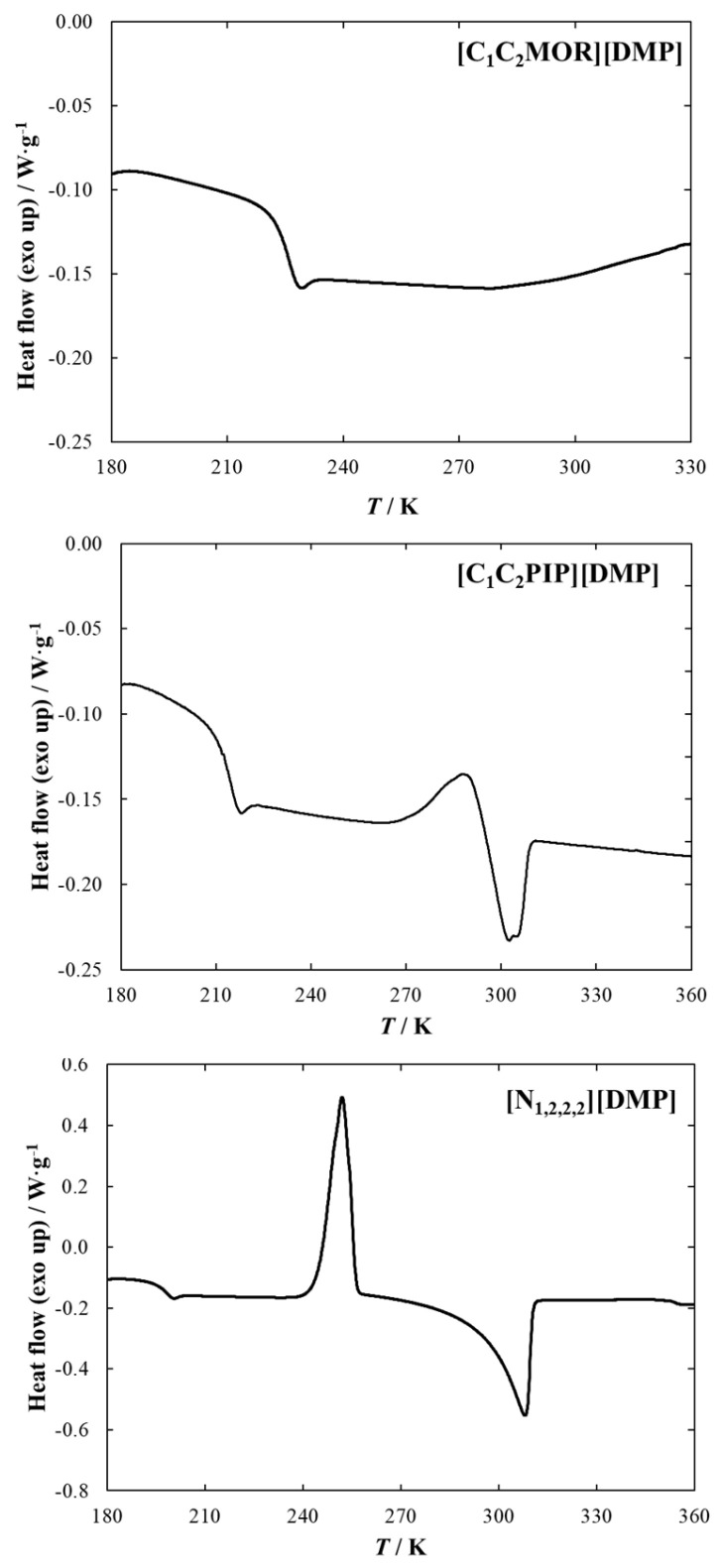
DSC diagrams (in heating mode) of pure iLs.

**Figure 2 molecules-28-01940-f002:**
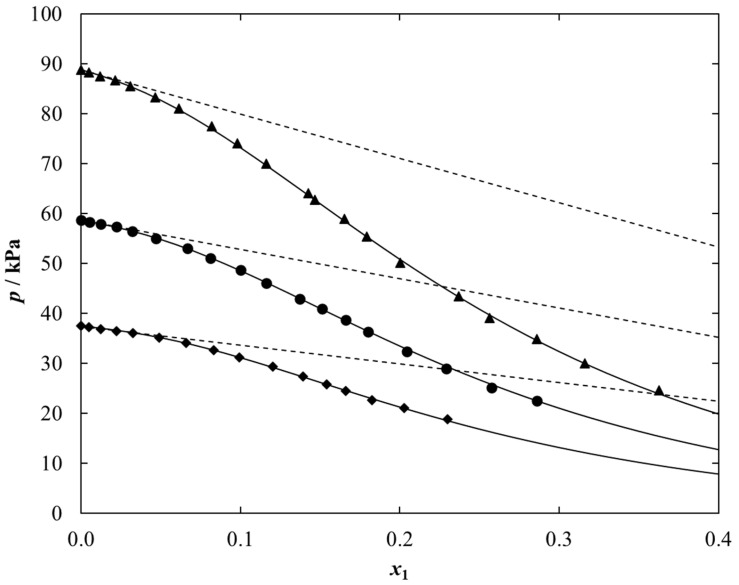
Plot of the experimental and calculated *P*–*x* data for {[C_1_C_2_MOR][DMP] (1) + Ethanol (2)} vs. ionic liquid mole fraction, *x*_1_, at different temperatures, *T*: ♦, 328.15 K; ●, 338.15 K; ▲, 348.15 K. Full points-experimental data; solid lines-NRTL equation with parameters given in Table 5; dashed lines-ideal solution.

**Figure 3 molecules-28-01940-f003:**
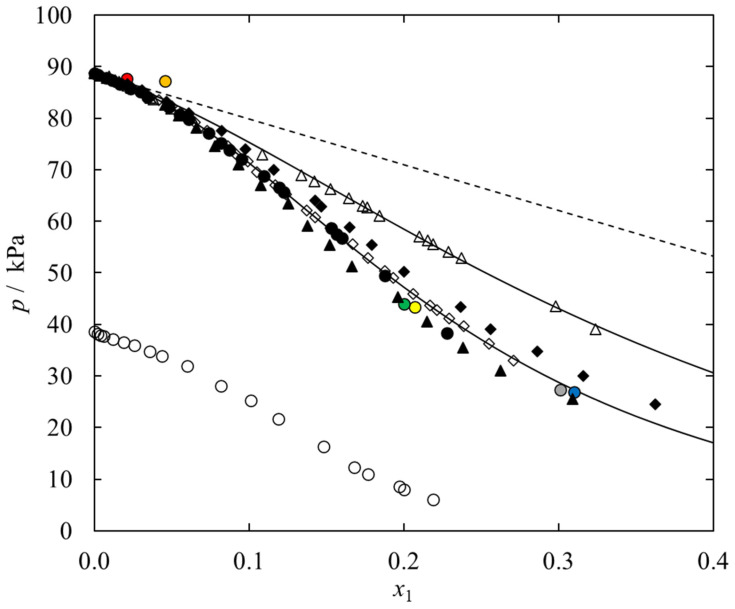
Comparison of the VLE data for the systems {[Cation][DMP] (1) + Ethanol (2)} vs. ionic liquid mole fraction, *x*_1_, at temperature *T* = 348.15 K; ♦, [C_1_C_2_MOR][DMP] (this work); ●, [C_1_C_2_PIP][DMP] (this work); ▲, [N_1,2,2,2_][DMP] (this work); ◊, [C_1_C_2_PYR][DMP] [31]; Δ, [C_1_C_2OH_PYR][DMP] [31]; ●, [C_1_C_1_IM][DMP] (*T* = 349.4 K) [35]; ●, [C_1_C_1_IM][DMP] (*T* = 347.75 K) [35]; ●, [C_1_C_2_IM][DMP] (*T* = 348 K) [38]; ●, [C_1_C_2_IM][DMP] (*T* = 347.45 K) [38]; ●, [N_1,2,2,2_][DMP] (*T* = 348.5 K) [40]; ●, [N_1,2,2,2_][DMP] (*T* = 349.29 K) [40]; ○, {LiBr (1) + Water (2)} [55]. Solid lines-NRTL equation with parameters given in Table 5; dashed lines-ideal solution.

**Figure 4 molecules-28-01940-f004:**
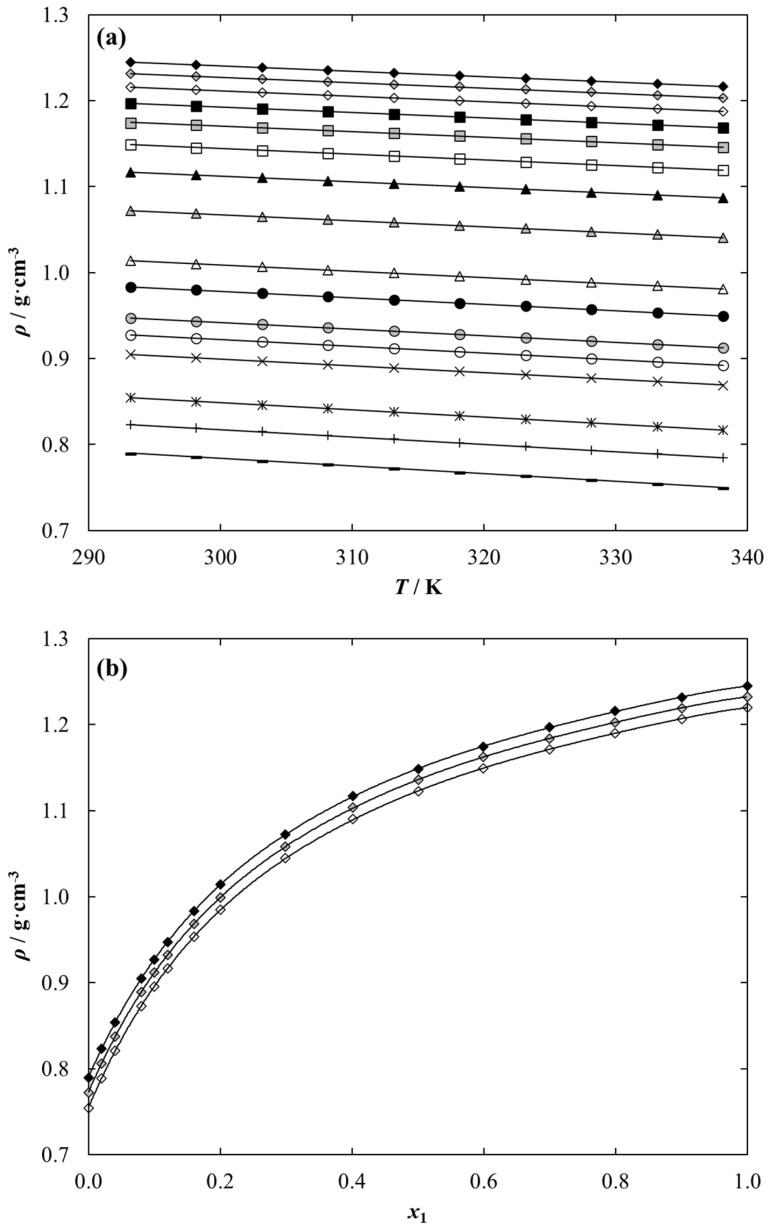
Temperature and composition dependence of liquid density data for {[C_1_C_2_MOR][DMP] (1) + Ethanol (2)} system as a function of (**a**) temperature for different composition, *x*_1_: ♦, 1.0000; ♦, 0.9012; ◊, 0.7990; ■, 0.6997; ■, 0.5994; □, 0.5012; ▲, 0.4012; ▲, 0.2991; Δ, 0.2005; ●, 0.1605; ●, 0.1200; ○, 0.1001; x, 0.0799; *, 0.0401; +, 0.0197; **–**, 0.0000. Points-experimental data; solid lines-correlation using Equation (10) with parameters given in Table 9. (**b**) Composition at different temperatures, *T*: ♦, 293.15 K; ♦, 313.15 K; ◊, 333.15 K. Points-experimental data; solid lines-correlation using Equation (12) with parameters given in Table 10.

**Figure 5 molecules-28-01940-f005:**
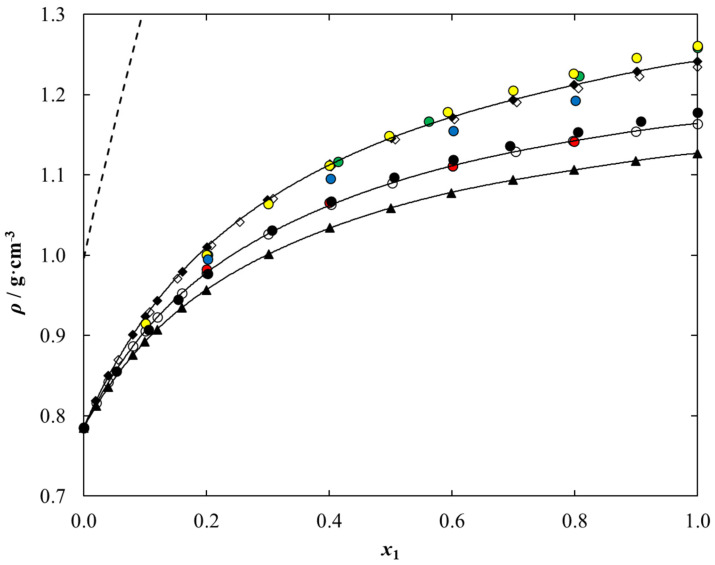
Experimental and calculated liquid density data of {Dimethyl Phosphate-Based IL (1) + Ethanol (2)} system vs. ionic liquid mole fraction, *x*_1_, at *T* = 298.15 K: ♦, [C_1_C_2_MOR][DMP] (this work); ○, [C_1_C_2_PIP][DMP] (this work); ▲, [N_1,2,2,2_][DMP] (this work); ●, [C_1_C_1_IM][DMP] [54]; ●, [C_1_C_1_IM][DMP] [35]; ●, [C_1_C_2_IM][DMP] [54]; ●, [C_1_C_4_IM][DMP] [49]; ●, [C_1_C_2_PYR][DMP] [41]; ◊, [C_1_C_2OH_PYR][DMP] [41]. Points-experimental, or literature density data; solid lines-calculated using Equation (12) with parameters given in Table 10; dashed lines-literature density data for {LiBr (1) + Water (2)} system [55].

**Figure 6 molecules-28-01940-f006:**
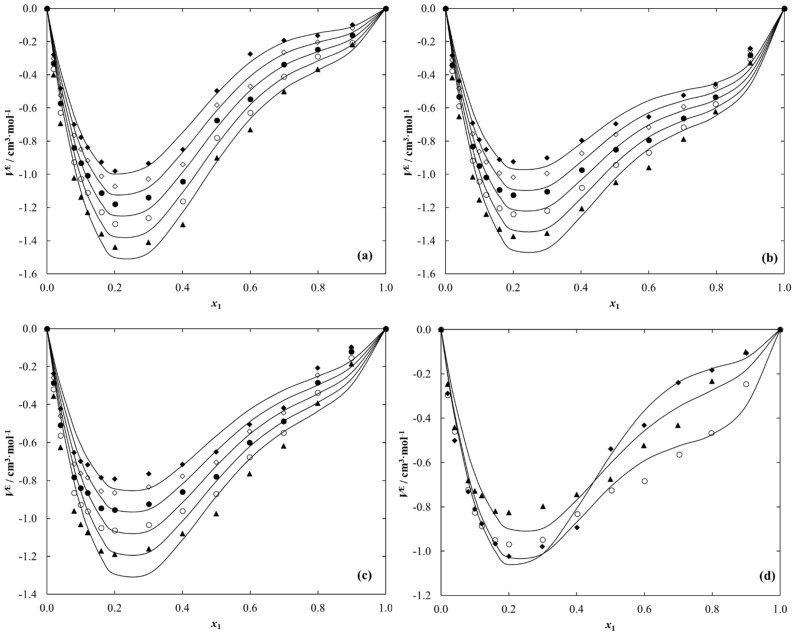
Excess molar volume, *V*^E^ for {IL (1) + Ethanol (2)}: (**a**) [C_1_C_2_MOR][DMP]; (**b**) [C_1_C_2_PIP][DMP]; (**c**) [N_1,2,2,2_][DMP] vs. ionic liquid mole fraction, *x*_1_, at different temperature, *T*: ♦, 293.15 K; ◊, 303.15 K; ●, 313.15 K; ○, 323.15 K; ▲, 333.15 K. (**d**) Comparison of the *V*^E^ at temperature *T* = 298.15 K: ♦, [C_1_C_2_MOR][DMP]; ○, [C_1_C_2_PIP] [DMP]; ▲, [N_1,2,2,2_][DMP]. Points-experimental results; solid lines-calculated using Equation (14) with parameters given in Table 11.

**Figure 7 molecules-28-01940-f007:**
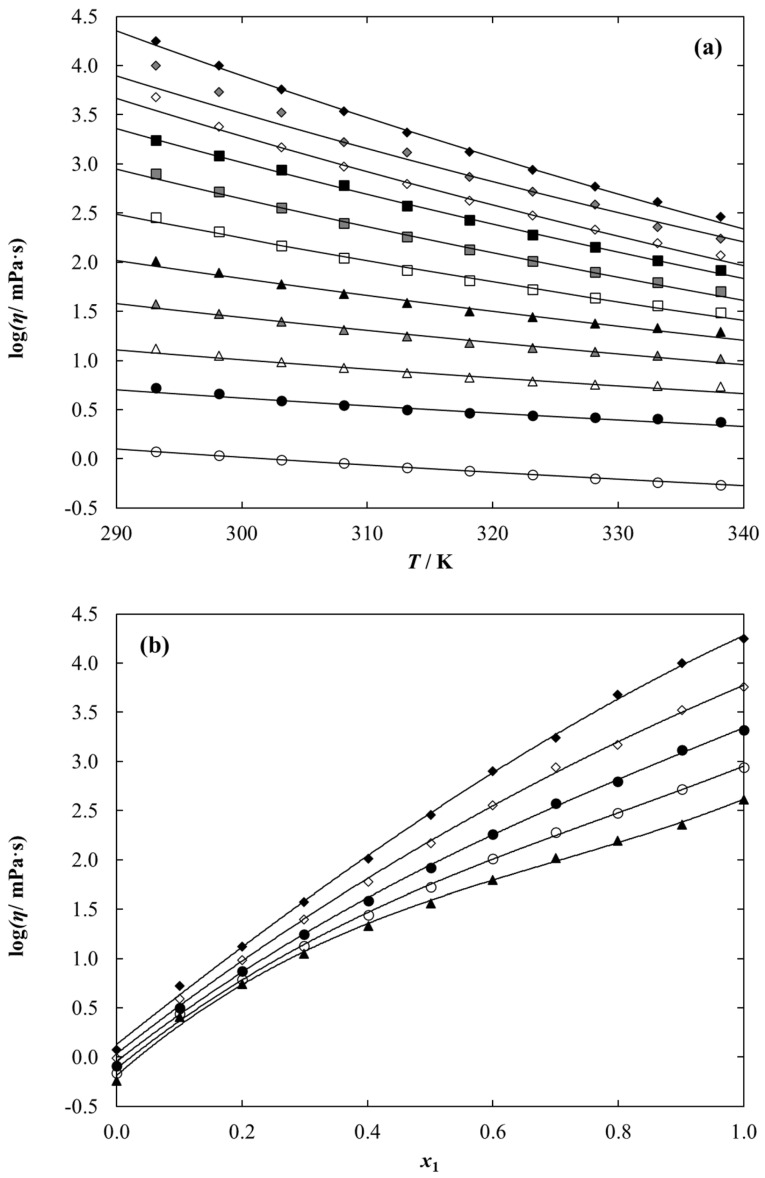
Experimental and calculated dynamic viscosity data for {[C1C2MOR][DMP] (1) + Ethanol (2)} binary system as a function of (**a**) temperature for different IL mole fraction, *x*_1_: ♦, 1.0000; ♦, 0.9012; ◊, 0.7990; ■, 0.6997; ■, 0.5994; □, 0.5012; ▲, 0.4012; ▲, 0.2991; Δ, 0.2005; ●, 0.1001; ○, 0.0000. Points-experimental data; solid lines-correlation using Equation (19) with parameters given in Table 12. (**b**) Composition at different temperature, T: ♦, 293.15 K; ◊, 303.15 K; ●, 313.15 K; ○, 323.15 K; ▲, 333.15 K. Points-experimental data; solid lines-guide to the eye.

**Figure 8 molecules-28-01940-f008:**
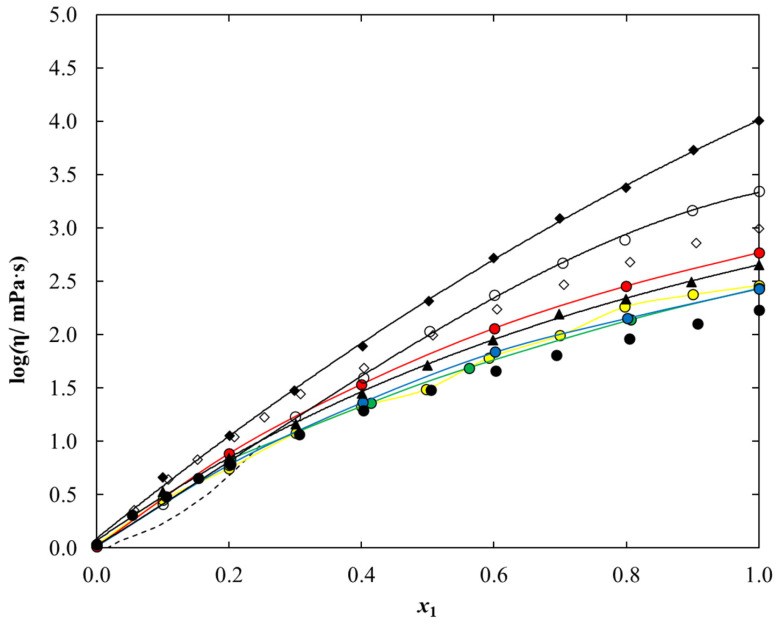
Experimental and calculated dynamic viscosity data of {Dimethyl Phosphate –Based IL (1) + Ethanol (2)} system vs. ionic liquid mole fraction, *x*_1_, at *T* = 298.15 K: ♦, [C_1_C_2_MOR][DMP] (this work); ○, [C_1_C_2_PIP][DMP] (this work); ▲, [N_1,2,2,2_][DMP] (this work); ●, [C_1_C_1_IM][DMP] [54]; ●, [C_1_C_1_IM][DMP] [35]; ●, [C_1_C_2_IM][DMP] [54]; ●, [C_1_C_4_IM][DMP] [49]; ●, [C_1_C_2_PYR][DMP] [31]; ◊, [C_1_C_2OH_PYR][DMP] [31]. Points-experimental, or literature density data; solid lines-calculated using Equation (19) with parameters given in Table 12; dashed lines-literature viscosity data for {LiBr (1) + Water (2)} system [55].

**Figure 9 molecules-28-01940-f009:**
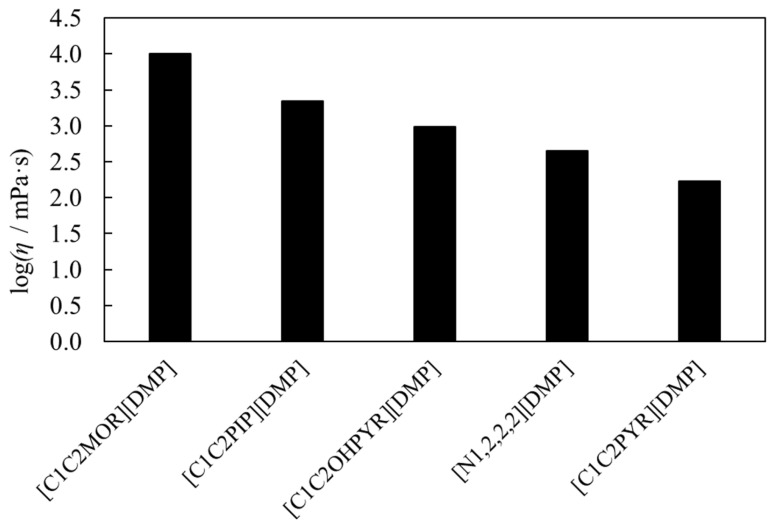
Dynamic viscosity value for pure dimethyl phosphate-based iLs.

**Table 1 molecules-28-01940-t001:** DSC Analysis for the dimethyl-phosphate-based iLs: glass transition temperature, *T*_g/_K, heat capacity at the glass transition temperature, Δ_g_*C*_p_/J·mol^−1^·K^−1^, temperature of (solid–solid) phase transition, *T*_tr,1_/K, enthalpy of (solid–solid) phase transition, Δ_tr,1_*H/*J·mol^−1^, melting temperature, *T*_m_/K and enthalpy of melting, Δ_m_*H*/J·mol^−1^ measured using the DSC technique at atmospheric pressure (*p* = 0.1 mPa) ^a^.

IL	*T*_g_/ (K)	Δ_g_*C*_p_/ (J mol^−1^ K^−1^)	*T*_tr,1_/ (K)	Δ_tr,1_*H*/ (kJ mol^−1^)	*T*_m_/ (K)	Δ_m_*H*/ (kJ mol^−1^)
[C_1_C_2_MOR][DMP] (this work)	222.2	132.7	–	–	–	–
[C_1_C_2_PIP][DMP] (this work)	211.0	103.9	–	–	295.2	2.026
[N_1,2,2,2_][DMP] (this work)	194.4	154.4	–	–	294.9	13.97
[C_1_C_2_PYR][DMP] [42]	189.7	119.6	274.5	0.861	285.2	0.727
[C_1_C_2OH_PYR][DMP] [42]	194.9 [42] 198.0 [51]	148.1 [42] 132.7 [51]	–	–	–	–

^a^ standard uncertainties *u* are as follows: *u*(*T*_g_) = 0.3 K; *u*(Δ_g_*C*_p_) = 3.3 J·mol^−1^ K^−1^; *u*(*p*) = 3 kPa.

**Table 2 molecules-28-01940-t002:** Experimental *P*–*T*–*x* data for the {[C_1_C_2_MOR][DMP] (1) + Ethanol (2)} binary system: equilibrium temperature, *T*/K; equilibrium pressure, *P*/kPa, ionic liquid mole fraction, *x*_1_; activity coefficient of ethanol, *γ*_2_
^a^.

*x* _1_	*P*/(kPa)	*γ* _2_	*x* _1_	*P*/(kPa)	*γ* _2_
*T* = 328.15 K
0.000	37.4	1.000	0.099	31.2	0.926
0.005	37.2	0.998	0.120	29.3	0.890
0.012	36.9	0.998	0.139	27.3	0.848
0.022	36.4	0.995	0.154	25.8	0.814
0.033	36.0	0.994	0.166	24.4	0.783
0.049	35.1	0.986	0.182	22.6	0.740
0.066	34.1	0.975	0.203	21.0	0.704
0.083	32.6	0.949	0.230	18.8	0.651
*T* = 338.15 K
0.000	58.6	1.000	0.116	46.1	0.890
0.005	58.3	0.999	0.137	42.9	0.849
0.012	57.8	0.998	0.151	40.9	0.821
0.022	57.3	0.999	0.166	38.6	0.789
0.032	56.4	0.993	0.180	36.3	0.754
0.047	54.9	0.983	0.204	32.3	0.692
0.067	53.0	0.969	0.229	29.0	0.641
0.081	51.1	0.948	0.257	25.2	0.578
0.100	48.7	0.923	0.286	22.5	0.536
*T* = 348.15 K
0.000	88.8	1.000	0.143	64.1	0.842
0.005	88.2	0.998	0.147	62.8	0.829
0.012	87.4	0.997	0.165	58.9	0.795
0.022	86.6	0.997	0.179	55.4	0.760
0.031	85.4	0.993	0.200	50.2	0.706
0.047	83.3	0.984	0.237	43.4	0.640
0.061	81.0	0.972	0.256	39.1	0.591
0.082	77.5	0.951	0.286	34.8	0.549
0.098	74.1	0.925	0.316	30.0	0.495
0.116	69.9	0.891	0.363	24.5	0.434

^a^ standard uncertainties *u* are as follows: *u*(*x*_1_) = 2·10^−3^; *u*(*P*) = 0.5 kPa and *u*(*T*) = 0.01 K.

**Table 3 molecules-28-01940-t003:** Experimental *P*–*T*–*x* data for the {[C_1_C_2_PIP][DMP] (1) + Ethanol (2)} binary system: equilibrium temperature, *T*/K; equilibrium pressure, *P*/kPa, ionic liquid mole fraction, *x*_1_; activity coefficient of ethanol, *γ*_2_
^a^.

*x* _1_	*P*/(kPa)	*γ* _2_	*x* _1_	*P*/(kPa)	*γ* _2_
*T* = 328.15 K
0.000	37.4	1.000	0.082	31.7	0.922
0.002	37.3	1.000	0.092	30.7	0.904
0.008	37.2	1.001	0.101	29.8	0.886
0.013	36.9	1.000	0.113	28.6	0.862
0.019	36.6	0.998	0.122	27.5	0.838
0.025	36.3	0.994	0.130	26.5	0.815
0.030	35.9	0.990	0.144	25.0	0.781
0.036	35.5	0.984	0.150	24.3	0.764
0.049	34.5	0.970	0.162	23.0	0.734
0.057	33.9	0.962	0.177	21.2	0.689
0.065	33.2	0.950	0.199	18.3	0.611
0.074	32.4	0.935			
*T* = 338.15 K
0.000	58.6	1.000	0.082	49.4	0.919
0.002	58.5	1.001	0.089	48.4	0.907
0.008	58.1	0.999	0.099	46.6	0.883
0.013	57.7	0.998	0.109	45.0	0.862
0.018	57.3	0.996	0.118	43.5	0.841
0.025	56.7	0.993	0.129	41.7	0.817
0.030	56.1	0.988	0.145	38.9	0.777
0.038	55.4	0.983	0.152	37.9	0.762
0.049	54.1	0.971	0.159	36.8	0.747
0.057	53.0	0.960	0.173	34.2	0.705
0.063	52.1	0.949	0.220	26.7	0.584
0.072	50.9	0.936			
*T* = 348.15 K
0.000	88.8	1.000	0.082	75.2	0.923
0.002	88.4	0.998	0.087	73.8	0.911
0.008	87.9	0.998	0.095	72.1	0.898
0.012	87.3	0.995	0.110	68.8	0.871
0.017	86.6	0.992	0.119	66.5	0.850
0.023	85.8	0.990	0.122	65.7	0.843
0.030	85.1	0.988	0.153	58.7	0.781
0.034	84.1	0.981	0.157	57.5	0.768
0.048	82.2	0.973	0.160	56.7	0.760
0.055	80.8	0.963	0.188	49.5	0.687
0.061	79.8	0.957	0.228	38.3	0.559
0.074	77.1	0.938			

^a^ standard uncertainties *u* are as follows: *u*(*x*_1_) = 2·10^−3^; *u*(*P*) = 0.5 kPa and *u*(*T*) = 0.01 K.

**Table 4 molecules-28-01940-t004:** Experimental *P–T–x* data for the {[N_1,2,2,2_][DMP] (1) + Ethanol (2)} binary system: equilibrium temperature, *T*/K; equilibrium pressure, *P*/kPa, ionic liquid mole fraction, *x*_1_; activity coefficient of ethanol, *γ*_2_ ^a^.

*x* _1_	*P*/(kPa)	*γ* _2_	*x* _1_	*P*/(kPa)	*γ* _2_
*T* = 328.15 K					
0.000	37.4	1.000	0.083	31.3	0.913
0.004	37.3	1.003	0.100	29.6	0.879
0.010	37.2	1.004	0.113	28.0	0.843
0.014	37.0	1.003	0.127	26.1	0.799
0.020	36.6	0.999	0.140	24.6	0.765
0.025	36.4	0.998	0.154	23.0	0.727
0.035	35.5	0.985	0.169	21.4	0.688
0.046	34.6	0.971	0.197	18.3	0.610
0.058	33.9	0.963	0.214	16.3	0.555
0.070	32.7	0.942			
*T* = 338.15 K					
0.000	58.6	1.000	0.098	46.6	0.881
0.004	58.5	1.001	0.112	44.0	0.846
0.009	58.1	1.000	0.126	41.3	0.806
0.014	57.7	0.998	0.139	38.7	0.766
0.019	57.3	0.996	0.154	36.2	0.729
0.024	56.8	0.993	0.167	33.7	0.690
0.035	55.8	0.987	0.197	29.4	0.624
0.047	54.4	0.973	0.214	26.5	0.575
0.057	53.0	0.957	0.238	22.7	0.508
0.068	51.3	0.939	0.258	20.4	0.468
0.082	48.9	0.909			
*T* = 348.15 K					
0.000	88.9	1.000	0.093	71.1	0.882
0.004	88.3	0.998	0.108	66.9	0.844
0.008	87.9	0.997	0.125	63.4	0.816
0.014	87.3	0.996	0.138	59.1	0.771
0.019	86.7	0.994	0.152	55.5	0.736
0.025	86.0	0.992	0.167	51.3	0.693
0.035	84.6	0.986	0.196	45.4	0.635
0.046	82.6	0.973	0.215	40.6	0.582
0.055	80.6	0.959	0.238	35.5	0.525
0.066	78.1	0.941	0.263	31.0	0.473
0.078	74.6	0.911	0.309	25.5	0.416

^a^ standard uncertainties *u* are as follows: *u*(*x*_1_) = 2·10^−3^; *u*(*P*) = 0.5 kPa and *u*(*T*) = 0.01 K.

**Table 5 molecules-28-01940-t005:** NRTL parameters fitted to experimental VLE data of {IL (1) + Ethanol (2)} with the average absolute deviation (AAD) and for the systems studied.

NRTL Parameters					%AAD
$\Delta g_{12}^{A}\;/J\cdot {\mathrm{mol}}^{-1}\cdot K^{-1}$	$\Delta g_{12}^{B}\;/J\cdot {\mathrm{mol}}^{-1}$	$\Delta g_{21}^{A}\;/J\cdot {\mathrm{mol}}^{-1}\cdot K^{-1}$	$\Delta g_{21}^{B}\;/J\cdot {\mathrm{mol}}^{-1}$	$\alpha_{12}=\alpha_{21}$	*P/*kPa
{[C_1_C_2_MOR][DMP] (1) + Ethanol (2)}					
−18.842	−4021.68	−35.693	28191.0	0.3	0.34
{[C_1_C_2_PIP][DMP] (1) + Ethanol (2)}					
−20.868	−4174.55	9.473	14494.33	0.3	0.45
{[N_1,2,2,2_][DMP] (1) + Ethanol (2)}					
−9.940	−8120.8	−139.231	64746.06	0.3	0.36

**Table 6 molecules-28-01940-t006:** Experimental values of density, *ρ*/g∙cm^−3^, excess molar volume, *V*^E^/cm^3^·mol^−1^, and dynamic viscosity, *η*/mPa·s for {[C_1_C_2_MOR][DMP] (1) + Ethanol (2)} binary mixture as a function of temperature, *T*/K and IL mole fraction, *x*_1_ at atmospheric pressure (*p* = 0.1 mPa) ^a^.

*T*/K	293.15	298.15	303.15	308.15	313.15	318.15	323.15	328.15	333.15	338.15
*x* _1_	*ρ*/g∙cm^−3^									
1.0000	1.24485	1.24171	1.23856	1.23542	1.23228	1.22916	1.22604	1.22292	1.21975	1.21670
0.9012	1.23170	1.22853	1.22537	1.22226	1.21921	1.21617	1.21307	1.20987	1.20665	1.20307
0.7990	1.21554	1.21237	1.20919	1.20601	1.20285	1.19961	1.19649	1.19339	1.19030	1.18720
0.6997	1.19667	1.19362	1.19038	1.18721	1.18406	1.18091	1.17774	1.17457	1.17141	1.16825
0.5994	1.17423	1.17198	1.16874	1.16549	1.16224	1.15899	1.15574	1.15250	1.14927	1.14605
0.5012	1.14860	1.14533	1.14203	1.13874	1.13544	1.13215	1.12887	1.12559	1.12232	1.11905
0.4012	1.11712	1.11373	1.11033	1.10694	1.10356	1.10019	1.09683	1.09347	1.09012	1.08677
0.2991	1.07234	1.06885	1.06534	1.06185	1.05836	1.05487	1.05139	1.04791	1.04443	1.04095
0.2005	1.01399	1.01035	1.00670	1.00305	0.99939	0.99574	0.99208	0.98842	0.98475	0.98108
0.1605	0.98349	0.97977	0.97604	0.97230	0.96856	0.96482	0.96107	0.95730	0.95353	0.94975
0.1200	0.94739	0.94359	0.93976	0.93592	0.93207	0.92822	0.92434	0.92046	0.91656	0.91263
0.1001	0.92727	0.92341	0.91953	0.91563	0.91173	0.90781	0.90387	0.89991	0.89593	0.89193
0.0799	0.90493	0.90102	0.89707	0.89312	0.88914	0.88515	0.88114	0.87710	0.87303	0.86893
0.0401	0.85415	0.85009	0.84599	0.84187	0.83773	0.83356	0.82936	0.82512	0.82084	0.81650
0.0197	0.82326	0.81910	0.81491	0.81068	0.80643	0.80214	0.79782	0.79344	0.78901	0.78453
0.0000	0.78953	0.78528	0.78096	0.77660	0.77221	0.76777	0.76328	0.75874	0.75413	0.74944
	*V*^E^/cm^3^·mol^−1^									
0.9012	−0.0993	−0.1064	−0.1172	−0.1349	−0.1625	−0.1895	−0.2077	−0.2106	−0.2187	−0.1509
0.7990	−0.1645	−0.1836	−0.2041	−0.2243	−0.2483	−0.2591	−0.2889	−0.3230	−0.3672	−0.3955
0.6997	−0.1922	−0.2390	−0.2635	−0.2979	−0.3363	−0.3744	−0.4116	−0.4507	−0.4998	−0.5374
0.5994	−0.2751	−0.4320	−0.4699	−0.5075	−0.5469	−0.5866	−0.6289	−0.6749	−0.7307	−0.7792
0.5012	−0.4964	−0.5388	−0.5820	−0.6280	−0.6750	−0.7244	−0.7781	−0.8350	−0.9015	−0.9622
0.4012	−0.8506	−0.8936	−0.9403	−0.9903	−1.0439	−1.1007	−1.1622	−1.2275	−1.3021	−1.3740
0.2991	−0.9330	−0.9788	−1.0279	−1.0816	−1.1382	−1.1978	−1.2625	−1.3315	−1.4086	−1.4858
0.2005	−0.9800	−1.0239	−1.0725	−1.1242	−1.1782	−1.2368	−1.2990	−1.3658	−1.4396	−1.5163
0.1605	−0.9257	−0.9667	−1.0125	−1.0608	−1.1122	−1.1676	−1.2268	−1.2890	−1.3588	−1.4316
0.1200	−0.8387	−0.8763	−0.9173	−0.9610	−1.0069	−1.0570	−1.1092	−1.1663	−1.2291	−1.2942
0.1001	−0.7763	−0.8104	−0.8486	−0.8888	−0.9319	−0.9777	−1.0266	−1.0786	−1.1362	−1.1974
0.0799	−0.6999	−0.7309	−0.7645	−0.8016	−0.8394	−0.8806	−0.9249	−0.9715	−1.0228	−1.0772
0.0401	−0.4815	−0.5010	−0.5230	−0.5470	−0.5722	−0.5993	−0.6285	−0.6591	−0.6931	−0.7285
0.0197	−0.2785	−0.2887	−0.3022	−0.3158	−0.3305	−0.3463	−0.3640	−0.3812	−0.4006	−0.4225
*η*/mPa·s										
1.0000	18870	10070	5752	3414	2093	1333	875.7	592.2	410.8	289.6
0.9012	5816	3534	2350	1658	1318	740.6	521.5	388.3	229.6	172.8
0.7990	3981	2398	1466	934.0	627.2	426.0	299.5	214.6	157.6	118.1
0.6997	1755	1223	866.5	608.0	376.0	268.0	190.9	142.1	104.6	82.77
0.5994	792.2	522.3	358.3	251.1	181.6	134.0	102.3	79.03	62.38	50.49
0.5012	285.4	205.2	148.3	110.4	83.74	65.41	53.11	43.2	36.04	30.87
0.4012	103.3	78.36	59.78	47.38	38.49	31.84	27.51	23.96	21.39	19.48
0.2991	37.30	30.04	24.79	20.6	17.57	15.14	13.45	12.35	11.22	10.47
0.2005	13.18	11.22	9.63	8.38	7.43	6.71	6.16	5.69	5.52	5.41
0.1001	5.27	4.59	3.92	3.51	3.17	2.92	2.78	2.65	2.56	2.38
0.0000	1.19	1.08	0.98	0.90	0.82	0.75	0.69	0.63	0.58	0.54

^a^ standard uncertainties *u* are as follows: *u*(*x*_1_) = 1·10^−4^; *u*(*ρ*) = 5·10^−3^ g cm^−3^; *u*_r_(*η*) = 0.03; *u*(*T*) = 0.01 K; *u*(*p*) = 3 kPa; *u*(*V*^E^) = 0.008 cm^3^·mol^−1^.

**Table 7 molecules-28-01940-t007:** Experimental values of density, *ρ*/g∙cm^−3^, excess molar volume, *V*^E^/cm^3^·mol^−1^, and dynamic viscosity, *η*/mPa·s for {[C_1_C_2_PIP][DMP] (1) + Ethanol (2)} binary mixture as a function of temperature, *T*/K and IL mole fraction, *x*_1_ at atmospheric pressure (*p* = 0.1 mPa) ^a^.

*T*/K	293.15 ^b^	298.15	303.15	308.15	313.15	318.15	323.15	328.15	333.15	338.15
*x* _1_	*ρ*/g∙cm^−3^									
1.0000	1.16712	1.16404	1.16094	1.15782	1.15465	1.15157	1.14852	1.14548	1.14242	1.13937
0.8993	1.15746	1.15434	1.15121	1.14802	1.14492	1.14184	1.13876	1.13567	1.13259	1.12952
0.7968	1.14571	1.14253	1.13930	1.13617	1.13306	1.12994	1.12682	1.12370	1.12060	1.11751
0.7031	1.13206	1.12901	1.12586	1.12269	1.11952	1.11635	1.11318	1.11004	1.10691	1.10379
0.6007	1.11454	1.11134	1.10812	1.10489	1.10166	1.09844	1.09524	1.09205	1.08888	1.08572
0.5028	1.09324	1.08994	1.08664	1.08335	1.08008	1.07682	1.07357	1.07033	1.06710	1.06388
0.4027	1.06633	1.06297	1.05961	1.05626	1.05291	1.04958	1.04626	1.04295	1.03965	1.03635
0.3001	1.03024	1.02685	1.02341	1.01997	1.01653	1.01310	1.00967	1.00625	1.00283	0.99942
0.2001	0.98145	0.97792	0.97436	0.97078	0.96720	0.96360	0.96001	0.95642	0.95282	0.94922
0.1597	0.95621	0.95255	0.94887	0.94520	0.94152	0.93784	0.93416	0.93046	0.92673	0.92299
0.1203	0.92693	0.92318	0.91941	0.91564	0.91186	0.90808	0.90429	0.90048	0.89666	0.89282
0.1001	0.90964	0.90583	0.90201	0.89818	0.89434	0.89049	0.88662	0.88274	0.87884	0.87492
0.0803	0.89065	0.88678	0.88289	0.87899	0.87508	0.87116	0.86721	0.86325	0.85926	0.85524
0.0402	0.84613	0.84214	0.83808	0.83400	0.82990	0.82577	0.82162	0.81743	0.81320	0.80892
0.0203	0.82031	0.81621	0.81204	0.80785	0.80363	0.79937	0.79508	0.79074	0.78636	0.78191
0.0000	0.78953	0.78528	0.78096	0.77660	0.77221	0.76777	0.76328	0.75874	0.75413	0.74944
	*V*^E^/cm^3^·mol^−1^									
0.8993	−0.2409	−0.2463	−0.2541	−0.2552	−0.2811	−0.2959	−0.3062	−0.3136	−0.3271	−0.3416
0.7968	−0.4575	−0.4662	−0.4711	−0.4963	−0.5332	−0.5561	−0.5758	−0.5952	−0.6227	−0.6522
0.7031	−0.5227	−0.5626	−0.5923	−0.6231	−0.6619	−0.6905	−0.7169	−0.7484	−0.7866	−0.8278
0.6007	−0.6526	−0.6831	−0.7158	−0.7514	−0.7943	−0.8308	−0.8690	−0.9100	−0.9594	−1.0128
0.5028	−0.6951	−0.7248	−0.7597	−0.8001	−0.8499	−0.8954	−0.9422	−0.9925	−1.0500	−1.1123
0.4027	−0.7960	−0.8320	−0.8735	−0.9205	−0.9738	−1.0259	−1.0805	−1.1391	−1.2052	−1.2760
0.3001	−0.9006	−0.9471	−0.9944	−1.0459	−1.1031	−1.1602	−1.2197	−1.2838	−1.3545	−1.4320
0.2001	−0.9233	−0.9689	−1.0178	−1.0692	−1.1255	−1.1809	−1.2405	−1.3044	−1.3741	−1.4504
0.1597	−0.9110	−0.9501	−0.9933	−1.0417	−1.0937	−1.1475	−1.2049	−1.2650	−1.3291	−1.3991
0.1203	−0.8506	−0.8864	−0.9265	−0.9707	−1.0182	−1.0683	−1.1215	−1.1775	−1.2394	−1.3066
0.1001	−0.7927	−0.8255	−0.8634	−0.9045	−0.9488	−0.9951	−1.0438	−1.0963	−1.1538	−1.2168
0.0803	−0.6922	−0.7216	−0.7552	−0.7920	−0.8317	−0.8737	−0.9175	−0.9650	−1.0167	−1.0731
0.0402	−0.4381	−0.4596	−0.4816	−0.5055	−0.5311	−0.5580	−0.5876	−0.6184	−0.6523	−0.6891
0.0203	−0.2836	−0.2968	−0.3102	−0.3255	−0.3413	−0.3579	−0.3762	−0.3948	−0.4164	−0.4391
	*η*/mPa·s									
1.0000	3276	2213	1485	1029	717.2	489.7	342.4	247.4	180.2	136.7
0.8993	2174	1473	1011	679.9	463.3	326.6	235.3	174.4	131.1	101.0
0.7968	1088	775.1	547.7	399.5	277.5	205.8	153.8	116.2	90.21	70.84
0.7031	662.0	471.0	330.4	235.7	172.9	129.7	99.70	77.71	61.75	49.88
0.6007	308.6	235.6	172.2	128.8	99.0	77.0	61.03	49.02	40.06	33.11
0.5028	135.7	108.4	83.4	65.52	52.1	42.3	34.63	28.72	24.06	20.59
0.4027	47.36	39.73	32.3	26.66	22.3	18.9	16.13	13.90	12.15	10.70
0.3001	19.86	17.06	14.5	12.48	10.8	9.4	8.27	7.29	6.45	5.90
0.2001	6.73	6.02	5.3	4.66	4.1	3.7	3.37	3.03	2.85	2.58
0.1001	2.85	2.59	2.3	2.09	1.9	1.7	1.58	1.45	1.34	1.27
0.0000	1.19	1.08	0.98	0.90	0.82	0.75	0.69	0.63	0.58	0.54

^a^ standard uncertainties *u* are as follows: *u*(*x*_1_) = 1·10^−4^; *u*(*ρ*) = 5·10^−3^ g cm^−3^; *u*_r_(*η*) = 0.03; *u*(*T*) = 0.01 K; *u*(*p*) = 3 kPa; *u*(*V*^E^) = 0.008 cm^3^·mol^−1^. ^b^ supercooled liquid.

**Table 8 molecules-28-01940-t008:** Experimental values of density, *ρ*/g∙cm^−3^, excess molar volume, *V*^E^/cm^3^·mol^−1^, and dynamic viscosity, *η*/mPa·s for {[N_1,2,2,2_][DMP] (1) + Ethanol (2)} binary mixture as a function of temperature, *T*/K and IL mole fraction, *x*_1_ at atmospheric pressure (*p* = 0.1 mPa) ^a^.

*T*/K	293.15 ^b^	298.15	303.15	308.15	313.15	318.15	323.15	328.15	333.15	338.15
*x* _1_	*ρ*/g∙cm^−3^									
1.0000	1.13000	1.12703	1.12405	1.12104	1.11800	1.11495	1.11191	1.10888	1.10585	1.10284
0.8987	1.12037	1.11735	1.11432	1.11127	1.10822	1.10519	1.10215	1.09912	1.09609	1.09306
0.7991	1.10938	1.10642	1.10336	1.10029	1.09722	1.09417	1.09111	1.08806	1.08500	1.08194
0.6988	1.09687	1.09374	1.09059	1.08745	1.08431	1.08119	1.07806	1.07495	1.07184	1.06874
0.5986	1.08088	1.07769	1.07448	1.07129	1.06810	1.06493	1.06176	1.05860	1.05545	1.05231
0.4998	1.06194	1.05871	1.05546	1.05222	1.04899	1.04576	1.04253	1.03931	1.03610	1.03289
0.4004	1.03727	1.03394	1.03061	1.02728	1.02396	1.02064	1.01733	1.01403	1.01074	1.00744
0.3011	1.00514	1.00171	0.99827	0.99484	0.99141	0.98799	0.98457	0.98116	0.97774	0.97433
0.1999	0.96064	0.95707	0.95349	0.94990	0.94631	0.94273	0.93914	0.93555	0.93196	0.92836
0.1600	0.93820	0.93455	0.93089	0.92723	0.92356	0.91989	0.91622	0.91253	0.90884	0.90514
0.1201	0.91144	0.90770	0.90394	0.90018	0.89641	0.89263	0.88885	0.88504	0.88123	0.87739
0.0998	0.89627	0.89248	0.88867	0.88485	0.88102	0.87717	0.87331	0.86944	0.86554	0.86162
0.0802	0.88003	0.87618	0.87231	0.86842	0.86453	0.86061	0.85668	0.85272	0.84874	0.84472
0.0402	0.84022	0.83621	0.83216	0.82810	0.82401	0.81989	0.81574	0.81156	0.80734	0.80306
0.0201	0.81643	0.81231	0.80814	0.80395	0.79974	0.79549	0.79119	0.78686	0.78247	0.77802
0.0000	0.78953	0.78528	0.78096	0.77660	0.77221	0.76777	0.76328	0.75874	0.75413	0.74944
	*V*^E^/cm^3^·mol^−1^									
0.8987	−0.0968	−0.1001	−0.1040	−0.1100	−0.1215	−0.1391	−0.1538	−0.1692	−0.1855	−0.1992
0.7991	−0.2078	−0.2336	−0.2458	−0.2620	−0.2835	−0.3113	−0.3370	−0.3643	−0.3915	−0.4175
0.6988	−0.4187	−0.4309	−0.4434	−0.4628	−0.4874	−0.5182	−0.5480	−0.5814	−0.6172	−0.6547
0.5986	−0.5048	−0.5222	−0.5406	−0.5670	−0.5985	−0.6363	−0.6754	−0.7172	−0.7638	−0.8131
0.4998	−0.6481	−0.6746	−0.7028	−0.7375	−0.7784	−0.8231	−0.8698	−0.9200	−0.9755	−1.0335
0.4004	−0.7140	−0.7430	−0.7766	−0.8152	−0.8596	−0.9083	−0.9608	−1.0173	−1.0798	−1.1449
0.3011	−0.7638	−0.7968	−0.8339	−0.8768	−0.9241	−0.9769	−1.0332	−1.0940	−1.1592	−1.2304
0.1999	−0.7911	−0.8258	−0.8651	−0.9080	−0.9550	−1.0076	−1.0632	−1.1230	−1.1888	−1.2597
0.1600	−0.7849	−0.8185	−0.8567	−0.8993	−0.9450	−0.9953	−1.0499	−1.1070	−1.1703	−1.2390
0.1201	−0.7174	−0.7486	−0.7838	−0.8232	−0.8654	−0.9115	−0.9620	−1.0143	−1.0729	−1.1356
0.0998	−0.6978	−0.7277	−0.7616	−0.7988	−0.8387	−0.8816	−0.9281	−0.9783	−1.0322	−1.0912
0.0802	−0.6532	−0.6805	−0.7120	−0.7459	−0.7831	−0.8226	−0.8657	−0.9108	−0.9605	−1.0138
0.0402	−0.4220	−0.4400	−0.4606	−0.4840	−0.5081	−0.5342	−0.5623	−0.5924	−0.6254	−0.6603
0.0201	−0.2362	−0.2464	−0.2583	−0.2720	−0.2869	−0.3029	−0.3191	−0.3372	−0.3564	−0.3776
	*η*/mPa·s									
1.0000	679.6	453.4	308.3	216.5	155.8	115.9	87.38	66.80	52.33	41.46
0.8987	456.3	311.7	217.0	156.6	115.0	86.57	66.31	51.39	40.64	32.77
0.7991	307.3	214.3	152.2	112.1	83.82	64.08	50.21	39.51	31.60	25.66
0.6988	197.2	156.8	115.5	84.00	64.38	50.80	40.12	31.19	25.37	20.88
0.5986	119.4	89.15	66.95	51.50	40.53	32.31	26.40	21.80	18.30	15.48
0.4998	67.67	51.89	40.71	32.43	26.40	21.70	18.21	15.45	13.17	11.50
0.4004	34.79	28.01	22.84	18.91	16.00	13.57	11.71	10.20	9.14	8.13
0.3011	17.41	14.58	12.43	10.65	9.24	8.15	7.20	6.45	5.88	5.39
0.1999	7.99	6.99	6.12	5.40	4.85	4.41	4.06	3.76	3.51	3.32
0.0998	3.77	3.38	3.01	2.71	2.46	2.27	2.12	1.98	1.80	1.65
0.0000	1.19	1.08	0.98	0.90	0.82	0.75	0.69	0.63	0.58	0.54

^a^ standard uncertainties *u* are as follows: *u*(*x*_1_) = 1·10^−4^; *u*(*ρ*) = 5·10^−3^ g cm^−3^; *u*_r_(*η*) = 0.03; *u*(*T*) = 0.01 K; *u*(*p*) = 3 kPa; *u*(*V*^E^) = 0.008 cm^3^·mol^−1^. ^b^ supercooled liquid.

**Table 9 molecules-28-01940-t009:** The parameters of Equations (10) and (11) for correlation temperature dependence of the density of {IL (1) + Ethanol (2)} binary systems ^a^.

*x* _1_	$\;\rho_{0}$	$10^{4}\alpha_{P}/K^{-1}$	10^4^·*σ*
{[C_1_C_2_MOR][DMP] (1) + Ethanol (2)}			
1.0000	1.24171	5.087	0.225
0.9012	1.22862	5.167	1.970
0.7990	1.21235	5.250	0.390
0.6997	1.19357	5.350	0.478
0.5994	1.17171	5.498	3.060
0.5012	1.14533	5.796	0.282
0.4012	1.11373	6.121	0.139
0.2991	1.06886	6.602	0.344
0.2005	1.01038	7.332	0.671
0.1605	0.97981	7.756	0.878
0.1200	0.94364	8.304	1.190
0.1001	0.92348	8.632	1.370
0.0799	0.90109	9.016	1.610
0.0401	0.85019	10.007	2.150
0.0197	0.81923	10.697	2.530
0.0000	0.78543	11.568	3.019
{[C_1_C_2_PIP][DMP] (1) + Ethanol (2)}			
1.0000	1.16401	5.354	0.332
0.8993	1.15432	5.430	0.235
0.7968	1.14251	5.533	0.271
0.7031	1.12899	5.638	0.521
0.6007	1.11133	5.830	0.183
0.5028	1.08993	6.050	0.059
0.4027	1.06297	6.338	0.063
0.3001	1.02685	6.757	0.426
0.2001	0.97795	7.426	0.811
0.1597	0.95259	7.851	0.932
0.1203	0.92323	8.328	1.031
0.1001	0.90589	8.644	1.223
0.0803	0.88685	9.009	1.424
0.0402	0.84223	9.990	2.095
0.0203	0.81632	10.646	2.498
0.0000	0.78565	11.572	3.019
{[N_1,2,2,2_][DMP] (1) + Ethanol (2)}			
1.0000	1.12706	5.418	0.625
0.8987	1.11736	5.487	0.315
0.7991	1.10640	5.576	0.533
0.6988	1.09374	5.776	0.149
0.5986	1.07768	5.955	0.747
0.4998	1.05871	6.166	0.231
0.4004	1.03395	6.485	0.210
0.3011	1.00172	6.918	0.341
0.1999	0.95710	7.596	0.635
0.1600	0.93459	7.970	0.826
0.1201	0.90775	8.457	1.079
0.0998	0.89254	8.758	1.292
0.0802	0.87625	9.094	1.512
0.0402	0.83631	10.043	2.095
0.0201	0.81243	10.696	2.495
0.0000	0.78699	11.537	3.019

^a^ standard uncertainties *u* are as follows: *u*(*x*_1_) = 1·10^−4^; *u*(*ρ*) = 5·10^−3^ g cm^−3^; *u*_r_(*η*) = 0.03; *u*(*T*) = 0.01 K; *u*(*p*) = 3 kPa; *u*(*V*^E^) = 0.008 cm^3^·mol^−1^.

**Table 10 molecules-28-01940-t010:** The parameters of Equations (12) and (13) for correlation composition dependence of the density of {IL (1) + Ethanol (2)} binary systems ^a^.

	$10^{2}a$	$a^{\prime }$	$10^{2}b$	$b^{\prime }$	$10^{2}c$	$c^{\prime }$	$10^{2}d$	$d^{\prime }$	$10^{2}e$	$e^{\prime }$	*σ*
[C_1_C_2_MOR][DMP] (1) + Ethanol (2)	−0.369	0.109	0.882	0.058	−0.937	0.018	0.487	0.073	−0.159	1.276	0.004
[C_1_C_2_PIP][DMP] (1) + Ethanol (2)	−0.229	−0.311	0.910	−0.174	−0.906	0.163	0.285	0.511	−0.105	1.105	0.004
[N_1,2,2,2_][DMP] (1) + Ethanol (2)	−0.307	0.119	0.694	0.076	−0.745	0.032	0.403	−0.028	−0.143	1.226	0.003

^a^ $\;\rho /\left(g\cdot {\mathrm{cm}}^{-3}\right)=\left(aT/K+a^{\prime }\right)x^{4}+\left(bT/K+b^{\prime }\right)x^{3}+\left(cT/K+c^{\prime }\right)x^{2}+\left(dT/K+d^{\prime }\right)x+\left(eT/K+e^{\prime }\right)$; $RMSE=\sqrt{\frac{\sum_{i=1}^{N}{\left(\rho^{calc}-\rho^{exp}\right)}^{2}}{N-P}}$.

**Table 11 molecules-28-01940-t011:** Parameters for the Redlich–Kister Equation Using for the Correlation of the Excess Molar Volumes, *V*^E^ for the {IL (1) + Ethanol (2)} Binary System. Along With the Standard Deviations, *σ*
^a^.

	10^2^·*a*_0_	10^2^·*a*_1_	10^2^·*a*_2_	*b* _0_	*b* _1_	*b* _2_	*σ*/cm^3^·mol^−1^
[C_1_C_2_MOR][DMP] (1) + Ethanol (2)	−4.121	3.644	−4.329	10.020	−6.258	8.474	0.058
[C_1_C_2_PIP][DMP] (1) + Ethanol (2)	−3.627	3.834	−4.850	7.978	−8.537	9.304	0.071
[N_1,2,2,2_][DMP] (1) + Ethanol (2)	−3.198	3.472	−4.904	7.122	−7.092	11.145	0.084

^a^ $V^{E}/({\mathrm{cm}}^{3}\cdot {\mathrm{mol}}^{-1})=x_{1}x_{2}\overset{k}{\underset{k=0}{\sum }}A_{k}{\left(x_{1}-x_{2}\right)}^{k}=x_{1}x_{2}\left[A_{0}+A_{1}\left(x_{1}-x_{2}\right)+A_{2}{\left(x_{1}-x_{2}\right)}^{2}\right]$; $A_{k}=a_{k}+b_{k}T/K$; $RMSE=\sqrt{\frac{\sum_{i=1}^{n}{\left(V_{i}^{E\left(\exp\right)}-V_{i}^{E\left(calc\right)}\right)}^{2}}{\left(n-p\right)}}$.

**Table 12 molecules-28-01940-t012:** The parameters of Equations (19) and (21) for correlation the temperature dependence of the dynamic viscosity of {IL (1) + Ethanol (2)} binary systems ^a^.

*x* _1_	*A*	10^−3^*B/*K	RMSE
{[C_1_C_2_MOR][DMP] (1) + Ethanol (2)}			
1.0000	−21.526	−9.150	
0.9012	−17.479	−7.670	
0.7990	−18.216	−7.732	
0.6997	−16.160	−6.929	
0.5994	−14.097	−6.056	0.091
0.5012	−11.247	−4.925	
0.4012	−8.077	−3.688	
0.2991	−6.062	−2.811	
0.2005	−4.407	−2.017	
0.1001	−4.223	−1.695	
0.0000	−5.519	−1.666	
{[C_1_C_2_PIP][DMP] (1) + Ethanol (2)}			
1.0000	−15.911	−7.034	
0.8993	−16.125	−6.977	
0.7968	−13.820	−6.098	
0.7031	−13.965	−5.995	
0.6007	−11.382	−5.014	6.073
0.5028	−9.439	−4.204	
0.4027	−7.433	−3.307	
0.3001	−6.238	−2.702	
0.2001	−5.338	−2.121	
0.1001	−5.155	−1.815	
0.0000	−5.502	−1.666	
{[N_1,2,2,2_][DMP] (1) + Ethanol (2)}			
1.0000	−14.534	−6.143	
0.8987	−13.720	−5.790	
0.7991	−12.935	−5.445	
0.6988	−11.951	−5.053	
0.5986	−10.640	−4.500	
0.4998	−9.152	−3.898	0.124
0.4004	−7.440	−3.204	
0.3011	−6.036	−2.593	
0.1999	−4.607	−1.946	
0.0998	−4.758	−1.778	
0.0000	−5.517	−1.666	

^a^ $\ln\left(\eta \right)=A-\frac{B}{T}$; $RMSE=\sqrt{\frac{\sum_{i=1}^{N}{\left(\eta^{calc}-\eta^{exp}\right)}^{2}}{N-P}}$.

**Table 13 molecules-28-01940-t013:** The structures and basic information of ILs under study: molecular weight (*M*), liquid density (*ρ*), and dynamic viscosity (*η*) at temperature *T* = 313.15 K and pressure *p* = 0.1 mPa ^a^.

Structure	Name, Abbreviation, CAS No.	*M*/ g·mol^−1^	*ρ*/ g·cm^−3^	*η*/ mPa·s
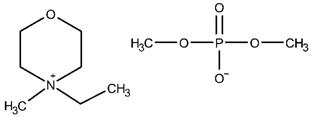	1-ethyl-1-methylmorpholinium dimethyl phosphate, [C_1_C_2_MOR][DMP]; CAS No. –	255.25	1.23228	2093
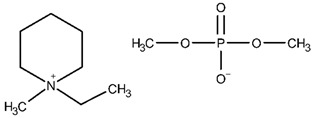	1-ethyl-1-methylpiperidinium dimethyl phosphate, [C_1_C_2_PIP][DMP]; CAS No. –	253.28	1.15465	717.2
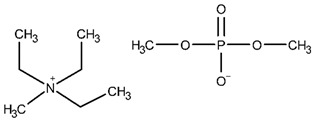	*N*,*N*,*N*-trimethyl-*N*-methylammonium dimethyl phosphate, [N_1,2,2,2_][DMP]; CAS No. –	241.26	1.11800	155.8

**Table 14 molecules-28-01940-t014:** Specifications of chemical samples.

Sample	Source	Water Content/ppm	Initial Mass Fraction Purity	Purification Method	Final Mass Fraction Purity	Analysis Method
[C_1_C_2_MOR][DMP]	Own synthesis	9000	−	Vacuum heating	≥0.950	Karl-Fischer, ^1^H NMR
[C_1_C_2_PIP][DMP]	Io-li-tec	7000	−	Vacuum heating extraction	≥0.970	Karl-Fischer, ^1^H NMR
[N_1,2,2,2_][DMP]	Own synthesis [41]	8000	−	Vacuum heating, extraction	≥0.950	Karl-Fischer, ^1^H NMR
Water	Own source	−	−	Reversed osmosis, ion exchange	0.9999	Density
Ethanol	POCH	90	0.998	−	0.998	−

## Data Availability

The data presented in this study are available in the article and Appendix A.

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
