# Peer review of "Investigation of Thermodynamic Properties of Dimethyl Phosphate-Based ILs for Use as Working Fluids in Absorption Refrigeration Technology"

_molecules, 2023, doi:10.3390/molecules28041940_

Round 1

Reviewer 1 Report

The authors described that the combination of basic studies on the effect of the cation structure of an ionic 33 liquid on the properties of their solutions with ethanol and the possibility of future application of 34 the tested systems in a viable refrigeration system.

The manuscript showed the details of all data to compare with thermodynamic and physicochemical properties.

However, the presentations of table 3,4,6,8 & 9, particularly, are written at length and hard to see them.

I recommend that the structure of the manuscript should be sorted out and to use supporting information section for clear presentation. 

The authors should described more clearly the reason of the sentense relationship below,

"The physicochemical properties reported here were compared to these data 585 for lithium bromide aqueous solution, conventionally used as a working fluid in absorp-586 tion refrigeration technology. The comparison shows that liquid density data for each bi-587 nary system is lower than those for {LiBr + water}."

Table 13 showed no literature data at all. This column should be omitted.

Author Response

  1. The authors described that the combination of basic studies on the effect of the cation structure of an ionic liquid on the properties of their solutions with ethanol and the possibility of future application of the tested systems in a viable refrigeration system. The manuscript showed the details of all data to compare with thermodynamic and physicochemical properties. However, the presentations of table 3,4,6,8&9, particularly, are written at length and hard to see them. I recommend that the structure of the manuscript should be sorted out and to use supporting information section for clear presentation.

The authors agree with the reviewer that the tables presented in the article are elaborate. The article contains an extensive amount of new experimental data. According to the authors, all the data presented in the mentioned tables are relevant and there is no possibility to modify the presentation of the experimental results. The only option is to move some of the tables to SM, which, according to the authors, will make it even more difficult to find the values contained there.

  1. The authors should described more clearly the reason of the sentense relationship below, "The physicochemical properties reported here were compared to these data for lithium bromide aqueous solution, conventionally used as a working fluid in absorption refrigeration technology. The comparison shows that liquid density data for each binary system is lower than those for {LiBr + water}."

The sentence indicated by the reviewer has been clarified. Such a relationship is related to the fact that the density of pure water is significantly higher than the density of ethanol proposed in this work as a circulating agent. Literature data shows that the density of pure LiBr is 3.46 g·cm-3 while the density of pure ionic liquids is nearly 3 times lower. This is primarily due to the structure of these salts. Lithium bromide is an inorganic salt composed of a small both cation and anion while an ionic liquid is composed of a large organic cation and a more extended anion.

  1. Table 13 showed no literature data at all. This column should be omitted.

The table has been revised.

Reviewer 2 Report

The paper is well-organized and includes new contributions with good merits for publication. The motivation of the work and the approach adopted are well. The manuscript is publishable after the below mentioned questions/comments are addressed.

Comments:

1) Title: It is suggested to change the title as follows:

Investigation of Thermodynamic Properties of dimethyl phosphate- based ILs for use as Working Fluids in Absorption Refrigeration Technology

2) Abstract: Each of the proposed binary system exhibit→ Each of the proposed binary systems exhibits

3) How much should be the density and viscosity of fluids for application in absorption refrigeration technology? Please bring the lowest values for density and viscosity of fluids for this application.

4) Why do the authors only investigate the effect of cation on the ionic liquid properties? What is the reason for chosen dimethyl phosphate anion in this study?

5) Recently, the deep eutectic solvents (DESs) have been introduced as new ionic liquids analogues. These green solvents have the feasibility of applying in various industrial applications, including absorption refrigeration systems (International Journal of Refrigeration Volume 113, May 2020, Pages 174-186). It would be great if you compare the properties obtained in this study with those of DESs reported in the literature.

Author Response

The paper is well-organized and includes new contributions with good merits for publication. The motivation of the work and the approach adopted are well. The manuscript is publishable after the below-mentioned questions/comments are addressed. Comments:

  • Title: It is suggested to change the title as follows: Investigation of Thermodynamic Properties of dimethyl phosphate- based ILs for use as Working Fluids in Absorption Refrigeration Technology

The title has been changed.

  • Abstract: Each of the proposed binary system exhibit→ Each of the proposed binary systems exhibits

It has been corrected in the manuscript.

  • How much should be the density and viscosity of fluids for application in absorption refrigeration technology? Please bring the lowest values for density and viscosity of fluids for this application.

In fact, there is no limit to the values ​​of viscosity and density. Both of these transport parameters should be as low as possible, which means that the absorption device will be smaller in size and less energy will be needed to transport the medium. The systems proposed by us show relatively low densities compared to, for example, the conventionally used system {LiBr + water}, which is shown in Figure 5, where we compare the densities of the systems proposed by us and the conventional system containing LiBr. The systems proposed by us, due to the presence of organic salt and ethanol, will have densities lower than the conventional system. In figure 8, in turn, we also presented a comparison of the systems proposed by us with the system {lithium bromide + water} for the viscosity in the range in which the salt does not crystallize. The systems proposed by us show higher viscosity values, but in the rich solution composition range, the viscosity values ​​are not high. In conclusion, there are no limits to the values ​​of viscosity and density. We compared these parameters for our systems in the figures with the most popular system used conventionally.

  • Why do the authors only investigate the effect of cation on the ionic liquid properties? What is the reason for chosen dimethyl phosphate anion in this study?

Research on liquids containing these anions was performed because it was noticed that the presence of these anions causes a significant decrease in the vapour pressure of systems with polar solvents. Such systems, therefore, should potentially exhibit high COP values. In addition, COP values ​​for water systems were previously calculated in our group (DOI:10.1016/j.fluid.2019.05.003) and it was observed that in this solvent, dimethyl phosphate anion-containing ILs exhibited high COP values.

  • Recently, the deep eutectic solvents (DESs) have been introduced as new ionic liquids analogues. These green solvents have the feasibility of applying in various industrial applications, including absorption refrigeration systems (International Journal of Refrigeration Volume 113, May 2020, Pages 174-186). It would be great if you compare the properties obtained in this study with those of DESs reported in the literature.

The comparison of ionic liquids with DESs as potential absorbents is really interesting. In the mentioned paper, the authors focused on the COP calculations for the {DES + NH3} systems. In our next work, we calculate the coefficients of performances for the systems {ILs + ethanol}.  taken from the literature and ones proposed by us. Then, having these values, we will be able to compare the COP values. It would be difficult to compare the VLE values ​​from the mentioned article with our data in this paper because the VLE measurements were made isothermally at different temperatures than ours.

Reviewer 3 Report

The paper is good, I have few comments: The title of the paper is so long, it should be reduced. The results presented in Table 2, 3, 4, 7 are not well discussed, as well the correlation of these results and figures.

Author Response

The paper is good, I have few comments:

  1. The title of the paper is so long, it should be reduced.

The title of this research has been shortened.

  1. The results presented in Table 2, 3, 4, 7 are not well discussed, as well the correlation of these results and figures.

The authors disagree with this comment. In our opinion, in the tables mentioned, the necessary quantities are presented, that is, the vapor pressure of the system under study as a function of the molar fraction of the ionic liquid, the pressure and temperature conditions are given, and the activity coefficients of water (γ2) are determined to allow the deviation of the system under study from ideality. The work presented here is a continuation of research conducted in this area for many years. The authors have experience both in the form of presenting this type of data, as well as conducting analysis and discussion of the data obtained. So far, the form of discussion of VLE data presented by us on several occasions, raised objections from other reviewers. (ex. Skonieczny, et. al.  J. Chem. Eng. Data 67, 4 (2022) 869–885; Królikowska, et. al. J. Chem. Eng. Data 66 (2021) 2281-2294; Królikowska, M., et. al.  J. Chem. Eng. Data 66 (2021) 8, 3300–3314; Królikowska, M., et al.  Fluid Phase Equilib. 547 (2021) 113175; Królikowska, M., et. al.  Thermochim. Acta. 671 (2019) 220 – 231; Zawadzki, M., et. al. J. Chem. Thermodyn. 98 (2016) 324337, or Królikowska, M., et. al.  J. Chem. Thermodyn. 70 (2014) 127137). In addition, it is worth mentioning that the presented work not only contains a comprehensive experimental data base for new systems not yet studied in the literature, it also contains valuable comparisons of experimental data with literature data for other ionic liquids, as well as with the {LiBr + water} system, commercially used as a refrigerant in absorption refrigeration technology.

According to the authors, the way the data are presented and their discussion taking into account the following aspects: (1) discussion of the values of the activity coefficients, and thus the type of deviations of the studied systems from perfection; (2) discussion of the influence of the IL structure, taking into account the effect of the alkyl chain length, the presence of heterocyclic cations, the presence of functional groups on the measured properties, and thus on intermolecular interactions; (3) correlation of the experimental VLE data, and (4) comparison of the experimental data with literature data for other phosphonium ionic liquids and the {LiBr + water} system should not be objectionable. According to the authors, all these aspects are extremely important for the purpose of the study.

Reviewer 4 Report

The article by Skonieczny M. et al. investigates the thermodynamic properties and physicochemical characteristics of binary systems based on ethanol and three ionic liquids. The authors synthesize three ionic liquids with different cation structures and measure their glass transition temperature, heat capacity, melting temperature, and melting enthalpy. After that, the authors obtain dozens of solutions of these ionic liquids in ethanol with different concentrations and estimate their density, viscosity, activity coefficient, equilibrium pressure, and other useful thermodynamic constants at different temperatures. In fact, this data is already valuable in itself, as it is referential information that can be of use to engineers and researchers. However, the author goes further and evaluates the obtained systems in terms of their applicability to refrigeration systems. The authors give their assessment of which of the ionic liquids and why is most suitable to be an absorbent in cooling circuits. In general, the work is very neat and written clearly, logically, and understandably in good English. The study is very detailed and comprehensive and yields useful data from both a scientific and practical point of view. In my opinion, the article can be published after several minor corrections.

Specific comments are as follows.

Line 112: A replacement is needed: “capacity, or transport” -> “capacity, and transport”.

Line 127: “but among them only in few the density, or viscosity” -> “, but only in few among them, there are data on the density or viscosity”.

Line 156, caption of Figure 1. It is necessary to specify the mode (heating or cooling) that was used.

Line 183: “and u(T) = 0.01 K.The vapor”. A new paragraph needs to start here.

Lines 287, 294, Table 5 (last column, second row): “?12 = ?12” -> “?12 = ?21”.

Lines 476-484: It is necessary to replace en-dash with the verbs "is" and "are" using the article "the".

Line 507: “Extremely high viscosity”. It is not an “extremely” high viscosity. The word “extremely” is unnecessary, "highest viscosity" is better.

Line 511: “using the following Andrade-type equation”. It is necessary to provide a reference to the Andrade equation (e.g., 10.3390/lubricants8050050). In addition, it should be pointed out that B=−Ea/R, where R is the gas constant and Ea is the flow activation energy. The flow activation energy has a physical meaning, in contrast to the meaningless fitting parameter B.

Figures 7, 8, and 9: These figures should be given in the semi-logarithmic form, i.e. log? dependences on T, x1, or ionic liquid composition. In linear coordinates, the viscosity values merge with the abscissa axis.

Line 578: “that the density values decrease in the following series: decreases in the following series”. Repeated phrase.

Line 711: “Dynsamic” -> “Dynamic”.

Line 731: Probably a replacement is needed: “between the cone and the plate” -> “between the cone truncation and the plate”.

Line 745. References. First line."Sun , J.; Fu, L; Zhang, S…". The reference is without the number, which apparently meant the number [0] from line 68.

Author Response

The article by Skonieczny M. et al. investigates the thermodynamic properties and physicochemical characteristics of binary systems based on ethanol and three ionic liquids. The authors synthesize three ionic liquids with different cation structures and measure their glass transition temperature, heat capacity, melting temperature, and melting enthalpy. After that, the authors obtain dozens of solutions of these ionic liquids in ethanol with different concentrations and estimate their density, viscosity, activity coefficient, equilibrium pressure, and other useful thermodynamic constants at different temperatures. In fact, this data is already valuable in itself, as it is referential information that can be of use to engineers and researchers. However, the author goes further and evaluates the obtained systems in terms of their applicability to refrigeration systems. The authors give their assessment of which of the ionic liquids and why is most suitable to be an absorbent in cooling circuits. In general, the work is very neat and written clearly, logically, and understandably in good English. The study is very detailed and comprehensive and yields useful data from both a scientific and practical point of view. In my opinion, the article can be published after several minor corrections.

Specific comments are as follows.

Line 112: A replacement is needed: “capacity, or transport” -> “capacity, and transport”.

Line 127: “but among them only in few the density, or viscosity” -> “, but only in few among them, there are data on the density or viscosity”.

Line 156, caption of Figure 1. It is necessary to specify the mode (heating or cooling) that was used.

Line 183: “and u(T) = 0.01 K.The vapor”. A new paragraph needs to start here.

Lines 287, 294, Table 5 (last column, second row): “?12 = ?12” -> “?12 = ?21”.

Lines 476-484: It is necessary to replace en-dash with the verbs "is" and "are" using the article "the".

Line 507: “Extremely high viscosity”. It is not an “extremely” high viscosity. The word “extremely” is unnecessary, "highest viscosity" is better.

Line 511: “using the following Andrade-type equation”. It is necessary to provide a reference to the Andrade equation (e.g., 10.3390/lubricants8050050). In addition, it should be pointed out that B=−Ea/R, where R is the gas constant and Ea is the flow activation energy. The flow activation energy has a physical meaning, in contrast to the meaningless fitting parameter B.

Figures 7, 8, and 9: These figures should be given in the semi-logarithmic form, i.e. log? dependences on T, x1, or ionic liquid composition. In linear coordinates, the viscosity values merge with the abscissa axis.

Line 578: “that the density values decrease in the following series: decreases in the following series”. Repeated phrase.

Line 711: “Dynsamic” -> “Dynamic”.

Line 731: Probably a replacement is needed: “between the cone and the plate” -> “between the cone truncation and the plate”.

Line 745. References. First line."Sun , J.; Fu, L; Zhang, S…". The reference is without the number, which apparently meant the number [0] from line 68.

All the reviewer's comments have been taken into account. Changes have been made to the manuscript.